# Anomaly matching in the symmetry broken phase: Domain walls, CPT, and the Smith isomorphism

**Itamar Hason[1⋆], Zohar Komargodski[2†] and Ryan Thorngren[3,4‡]**

**1** Raymond and Beverly Sackler School of Physics and Astronomy, Tel-Aviv University, Israel
**2** Simons Center for Geometry and Physics, SUNY, Stony Brook, NY, USA
**3** Department of Condensed Matter Physics, Weizmann Institute of Science, Israel
**4** Center for Mathematical Sciences and Applications,
Harvard University, Cambridge, MA, USA

⋆ itamarhason@gmail.com, † zkomargodski@scgp.stonybrook.edu,
‡ ryan.thorngren@cmsa.fas.harvard.edu

## Abstract

Symmetries in Quantum Field Theory may have 't Hooft anomalies. If the symmetry is unbroken in the vacuum, the anomaly implies a nontrivial low-energy limit, such as gapless modes or a topological field theory. If the symmetry is spontaneously broken, for the continuous case, the anomaly implies low-energy theorems about certain couplings of the Goldstone modes. Here we study the case of spontaneously broken discrete symmetries, such as $\mathbb{Z}_2$ and $T$. Symmetry breaking leads to domain walls, and the physics of the domain walls is constrained by the anomaly. We investigate how the physics of the domain walls leads to a matching of the original discrete anomaly. We analyze the symmetry structure on the domain wall, which requires a careful analysis of some properties of the unbreakable $CPT$ symmetry. We demonstrate the general results on some examples and we explain in detail the mod 4 periodic structure that arises in the $\mathbb{Z}_2$ and $T$ case. This gives a physical interpretation for the Smith isomorphism, which we also extend to more general abelian groups. We show that via symmetry breaking and the analysis of the physics on the wall, the computations of certain discrete anomalies are greatly simplified. Using these results we perform new consistency checks on the infrared phases of $2+1$ dimensional QCD.


# 1 Introduction

Suppose a quantum field theory in $d+1$ space-time dimensions has a spontaneously broken $\mathbb{Z}_2$ symmetry generated by the unitary operator $U$. Such a theory has two degenerate ground states related by $U$. In this situation the theory admits a protected dynamical excitation that interpolates between these two vacua, known as a domain wall. It may be analyzed by choosing a coordinate $x_\perp$ and frustrated boundary conditions for $x_\perp \to \pm\infty$ which force the system into one of the ground states for $x_\perp \to \infty$ and its $U$-conjugated ground state for $x_\perp \to -\infty$.

The domain wall always admits massless Nambu-Goldstone bosons due to the spontaneously broken translational symmetry in the normal coordinate $x_\perp$, with action given by the Nambu-Goto theory, but in many cases there could be other parametrically light excitations trapped on the domain wall, such as the Jackiw-Rebbi modes we discuss below.

The following basic question is the starting point of this paper: Does the original $\mathbb{Z}_2$ symmetry act on the Hilbert space of the domain wall? This question is sharply defined when there are light excitations (relative to the bulk excitations) trapped on the wall beyond the obvious translational Nambu-Goldstone modes, but the question makes sense also in various other situations which we will discuss below.

On the one hand, it would seem that the answer is positive since intuitively the $\mathbb{Z}_2$ symmetry is restored on the wall, since the domain wall is localized where the order parameter for the $\mathbb{Z}_2$ symmetry vanishes. On the other hand, the answer seems to be negative since the $\mathbb{Z}_2$

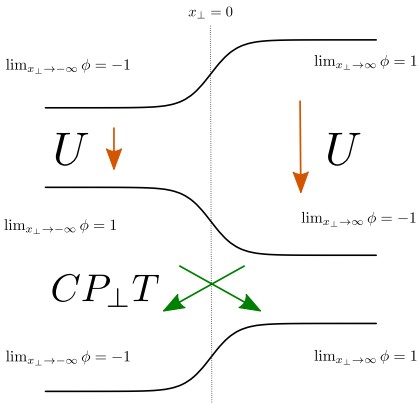

Figure 1: The $\mathbb{Z}_2$ domain wall is created by imposing frustrated boundary conditions for the order parameter $\phi$ along a coordinate $x_\perp$ or breaking the symmetry with a spatially-varying potential. Indeed, the global symmetry $U$ does not act on the wall but if we combine it with $CP_\perp T$, a canonical symmetry which involves a reflection in the normal coordinate $x_\perp$, then we obtain a symmetry of the domain wall degrees of freedom, which is anti-unitary if $U$ is unitary and vice versa.

transformation does not leave the boundary conditions at $x_\perp \to \pm\infty$ invariant—it exchanges the two sides of the wall—and so the $\mathbb{Z}_2$ symmetry cannot be considered a symmetry.

It turns out that neither of the options above is entirely correct. The resolution of this general question proceeds as follows. Consider acting with the spontaneously broken $\mathbb{Z}_2$ operator $U$. This interchanges the two vacua on either side of the wall and hence does not leave the bulk invariant. We then want to apply some "canonical" reflection symmetry across the wall to restore the boundary conditions (it is canonical in the sense that it always exists; for instance, parity symmetry does not always exist). We obtain such a canonical symmetry from Lorentz invariance, using the $CPT$ theorem, which guarantees us some space-orientation-reversing symmetry, which by combination with a rotation can be chosen to act by reflection in the $x_\perp$ coordinate. To emphasize this, we write it $CP_\perp T$.

We therefore consider the symmetry

$$T' = U \cdot (CP_\perp T)\,, \tag{1}$$

which by construction leaves the boundary conditions invariant and therefore indeed acts on the wall! But observe that this is an anti-unitary symmetry, since $CP_\perp T$ is anti-unitary. Thus, the correct answer to the question about the fate of the spontaneously broken, unitary, $\mathbb{Z}_2$ symmetry is that it becomes some anti-unitary symmetry acting on the wall.

By the same argument, if we began with a spontaneously broken anti-unitary $\mathbb{Z}_2$ symmetry, the domain wall would inherit a unitary symmetry. In fact, we will derive a 4-periodic dimensional hierarchy for $\mathbb{Z}_2$:

$$\begin{Bmatrix} \text{unitary} \\ U^2=1 \end{Bmatrix} \implies \begin{Bmatrix} \text{anti}-\text{unitary} \\ T^2=1 \end{Bmatrix} \implies \begin{Bmatrix} \text{unitary} \\ U^2=(-1)^F \end{Bmatrix} \implies \begin{Bmatrix} \text{anti}-\text{unitary} \\ T^2=(-1)^F \end{Bmatrix} \implies \begin{Bmatrix} \text{unitary} \\ U^2=1 \end{Bmatrix}, \tag{2}$$

where $(-1)^F$ is the fermion parity operator and the arrow indicates the induced symmetry on the domain wall. For bosons the hierarchy is 2-periodic, obtained from the one above by removing $(-1)^F$:

$$\begin{Bmatrix} \text{unitary} \\ U^2=1 \end{Bmatrix} \implies \begin{Bmatrix} \text{anti}-\text{unitary} \\ T^2=1 \end{Bmatrix} \implies \begin{Bmatrix} \text{unitary} \\ U^2=1 \end{Bmatrix}\,. \tag{3}$$

We will also explore similar hierarchies for larger symmetry groups and domain wall junctions.

With this understanding of the structure of the symmetries at the domain wall or junction, we can ask interesting questions about 't Hooft anomalies. For instance, the original theory determines the dynamics of the domain walls and junctions, but suppose we know the anomalies on the domain walls and junctions, what can we infer about the anomalies of the original theory?

We will show that for $\mathbb{Z}_2$ symmetries of all four types above, the anomaly on the domain wall determines the original anomaly. The proof of this extends the so-called Smith isomorphism theorem of cobordism theory. For more general symmetry groups, one typically has to study multiple types of domain walls and junctions to obtain the anomaly. However we will also show that in some cases the anomaly on the wall is not uniquely determined by the original anomaly, and with different symmetry breaking potentials there could be different anomalies.

In general it is hard to compute the discrete time reversal or $\mathbb{Z}_2$ anomalies of an interacting theory. But by repeatedly using the anomaly-matching relations we derive for the domain walls, we can reduce the calculation of the anomaly either to a gravitational anomaly, or all the way down to quantum mechanics, where the computation of the anomaly just amounts to determining the projective representation of the symmetry group on the ground states.

To put this discussion in context, recall that for continuous symmetries with a local 't Hooft anomaly there are essentially two logical options:

- The vacuum is invariant under the symmetry: In this case there must be some massless modes on which the unbroken symmetry acts (see [44] and references therein). We should think about this as a conformal field theory which may or may not be trivial.

- The symmetry is broken spontaneously: There are massless Nambu-Goldstone modes corresponding to the broken symmetries. The anomaly leads to various interactions among these Nambu-Goldstone modes in conjunction with prescribed couplings to background fields which lead to tree-level diagrams that reproduce the anomaly [45, 46].

For discrete symmetries the story is less clear and this paper is merely a step in that direction. It is still true that there are essentially two options, corresponding to an invariant vacuum or symmetry breaking. In the former case, some (but not all) discrete anomalies can be reproduced by a topological field theory—massless modes are not always necessary (see, for example, [19, 21–24, 47]). In the symmetry breaking case, there are no Nambu-Goldstone bosons but there are domain walls instead. So this paper is essentially about how these domain walls reproduce the original anomaly. This is the discrete avatar of the question about how Nambu-Goldstone bosons reproduce continuous anomalies. What we find is that for the simplest possible symmetry classes (essentially those in (2),(3)) the domain wall worldvolume theory *itself* has to have an anomaly and therefore must support multiple vacua, massless particles, or a topological theory.

For instance, in the fermionic case, we will argue for a general formula, relating the time reversal $T^2 = (-1)^F$ anomaly in $2 + 1$ dimensions, $\nu_3$ (which is defined mod 16) and a $\mathbb{Z}_2$ anomaly on its $1 + 1$ dimensional domain wall, $\nu_2$ (which is defined mod 8):

$$\nu_3 = 2\nu_2 - 2(c_R - c_L) \mod 16, \tag{4}$$

where $c_L$ and $c_R$ are the left and right central charges of the theory on the domain wall, respectively. The result (4) applies when the theory on the wall does not break the $\mathbb{Z}_2$ symmetry spontaneously. In the event that it does, there is a further reduction to quantum mechanics and the matching of the anomalies is even simpler. We will discuss the details of the case of a nonsymmetric vacuum on the wall in the main text. The formula (4) allows us to extract the original time reversal anomaly in 2+1 dimensions from domain wall constructions. It is

typically much easier to compute the discrete $\mathbb{Z}_2$ anomaly in 1+1 dimensions and the central charges are likewise straightforward to compute. Interestingly, as we change the coupling constants of a given 2+1 dimensional theory, different domain walls may appear with different $\nu_2, c_L, c_R$. But due to (4) the combination on the right hand side is always the same.

In the bosonic case, consider for instance a bosonic theory in 1+1 dimensions (say, free of gravitational anomalies). It may have a $\mathbb{Z}_2$ symmetry with a 't Hooft anomaly. If the symmetry breaks spontaneously then there is a domain wall which is a kink, essentially a point particle for a low-energy observer. The anomaly implies that time reversal symmetry acts projectively on this particle, meaning with $T^2 = -1$ on the Hilbert space, so there is an exact Kramers degeneracy over the entire spectrum of such kinks in the system with frustrated boundary conditions:

$$\mathbb{Z}_2 \text{ anomaly in } 1+1 \text{ dimensions} \longrightarrow \text{Kramers doublet domain walls (kinks) .} \qquad (5)$$

To demonstrate these general ideas we consider three classes of examples:

- Fermions in $2+1$ dimensions. Such theories have a time reversal anomaly classified by $\mathbb{Z}_{16}$. For free fermions, one can compute it directly in $2+1$ dimensions by carefully studying the Dirac operator on unorientable space-times [34]. This is quite delicate. We instead compute the anomalies by coupling the theory to a heavy pseudo-scalar which we condense. This reduces the problem to a more familiar problem in $1+1$ dimensions and we can further reduce it to quantum mechanics by studying the domain wall within the domain wall. This example also demonstrates that there can be multiple domain walls with different anomalies depending on the details of the symmetry breaking, but all of their anomalies must match the original theory.

- Abelian gauge theory in $1+1$ dimensions. We discuss a particular $\mathbb{Z}_2$ symmetry in the $\mathbb{CP}^1$ model and show that it has an anomaly by reducing the problem to the quantum mechanics on the domain wall in a spontaneously broken phase. The domain walls form a Kramers doublet demonstrating (5). We show how this surprising $\mathbb{Z}_2$ anomaly is consistent with the deformations of the theory.

- Sigma models with a Wess-Zumino term or Hopf term in $2+1$ dimensions. Such theories appear in the infrared of interesting systems such as $2+1$ dimensional gauge theories. These models often have nontrivial anomalies involving time reversal symmetry. We study the symmetries and domain walls of these models. Some of our results provide new consistency checks of conjectured renormalization group flows in $2+1$ dimensional QCD.

The outline of the paper is as follows. In Section 2 we discuss the properties of the $CPT$ symmetry and use it to derive the dimensional hierarchy for $\mathbb{Z}_2$ symmetries, as well as derive an anomaly-matching condition (by anomaly-matching we mean the relationship between the domain wall anomaly and the anomaly of the original theory). In Section 3 we discuss several examples in detail. In Section 4 we give a mathematical perspective on the anomaly matching condition based on cobordism theory, including proofs of the Smith isomorphism and some generalizations.

*Note Added: As this paper was being completed, we were made aware of a related work [51] which studies the reduction from the unitary $U^2 = 1$ to anti-unitary $T^2 = 1$ case of (2), captured by the classic Smith isomorphism of Section 4.4. Some related calculations also appear in [3].*

## 2 $CPT$ and the Domain Wall Symmetry Algebra

### 2.1 A Canonical $CPT$ and its Properties

A generic QFT does not necessarily have a time reversal symmetry, parity, or any unitary global symmetry. But in a unitary QFT with Lorentz invariance there is always an anti-unitary symmetry called $CPT$ which reverses time ($T$), a spatial coordinate ($P$), and may act on internal degrees of freedom ($C$). This $CPT$ symmetry is not unique, of course. For one, we can conjugate by a spatial rotation symmetry to obtain a $CPT$ which involves reflection around a different spatial coordinate. Also, if the theory admits a unitary internal symmetry, $U$, we can consider $U \cdot CPT$, which is another $CPT$-like symmetry.

In spelling out some properties of the $CPT$ symmetry below, we have to be precise about which $CPT$ symmetry we have in mind. One constructive way to think about it is that in any given relativistic QFT there is one (up to rotations) *canonical $CPT$* symmetry which is obtained in the following way. We first deform the theory by arbitrary Lorentz-invariant perturbations. This is guaranteed to break all the internal global symmetries, but there is still one unbreakable $CPT$ symmetry that survives. This is our *canonical $CPT$* symmetry to which the statements below pertain.[1] Of course, once we have identified this canonical $CPT$ symmetry we can use it in the original, undeformed theory, which possibly has various other symmetries.

There is one subtlety (other than the rotation degree of freedom which we fix when we set the direction of $P$) in the definition of $CPT$ through the procedure above, which is that there is always an unbreakable internal unitary symmetry $(-1)^F$ (where $F$ is the fermion number) which is part of the Lorentz group. So we can still combine $CPT$ with $(-1)^F$ if we wish. However, that will not make any difference for the statements below.

There are several important properties of the canonical $CPT$ symmetry (from now on we often omit the word 'canonical'):

1. Any unitary internal symmetry $U$ commutes with $CPT$:

$$U \cdot (CPT) = (CPT) \cdot U \,. \tag{6}$$

   This follows in essence from the Coleman-Mandula theorem. (This also applies for $U = (-1)^F$.)

2. Any time reversal symmetry $T$ commutes with $CPT$ up to the fermion parity:

$$T \cdot (CPT) = (-1)^F (CPT) \cdot T \,. \tag{7}$$

   The proof is given in Appendix A. Equation (7) means that on bosonic states $T$ and $CPT$ commute while on fermionic states they anti-commute.

3. 
$$(CPT)^2 = 1 \,. \tag{8}$$

   The proof of this proceeds as follows: if the right hand side were nonzero and not a c-number it would have to be a unitary non-space time symmetry, which would be in contradiction with the $CPT$ symmetry being canonical, except if it were $(-1)^F$ which is also unbreakable. It is also easy to rule out the option of a pure c-number on the right hand side of (8): We cannot absorb this c-number in the definition of $CPT$ since it is anti-unitary and hence $(e^{i\alpha}CPT)^2 = (CPT)^2$. But it suffices to assume that the ground state is $CPT$ invariant to arrive at $(CPT)^2 = 1$ since if the ground state is invariant

---

[1] In this discussion we ignore higher-form symmetries, which cannot be broken by local perturbations of the Lagrangian, but these do not introduce any ambiguity into the definition of $CPT$.

$CPT|0\rangle$ must give $e^{i\alpha}|0\rangle$ for some $\alpha$ and hence acting on it again and using the anti-unitary nature of $CPT$ we find $(CPT)^2|0\rangle = 1$. Now, since $(CPT)^2$ is assumed to be a c-number it must be 1 in all the states and not just the vacuum. Thus it only remains to decide whether $(CPT)^2 = (-1)^F$ or $(CPT)^2 = 1$. We show in Appendix A that the right answer is (8).[2]

An elegant and nontrivial consistency check of (6),(7), and (8) is that these relations are compatible with domain wall constructions, meaning that the symmetries on the wall we obtain from combining with $CP_\perp T$ continue to satisfy the claimed commutation relations with the $CPT$ intrinsic to the wall. Let us see how this comes about.

We start from a $d+1$ dimensional QFT with time reversal symmetry, $T$. We assume it is spontaneously broken and hence there are two different vacua related by $T$. We then consider the domain wall between these two vacua. As explained in the introduction we can consider $U = T \cdot CPT$ which is a symmetry that leaves the bulk invariant if the operator $P$ is taken to act perpendicularly to the wall.

Since $U$ is unitary and does not act on the space-time of the wall, it should commute with the $CPT$ symmetry of the wall which we will denote by $(CPT)_d$, while $(CPT)_{d+1}$ will be reserved for the original $CPT$ in the theory. $(CPT)_{d+1}$ and $(CPT)_d$ can be related by a conjugation by a $\pi/2$ spatial rotation in the plane that includes the vector perpendicular to the wall and a vector on the wall, so $(CPT)_d = R(-\pi/2) \cdot (CPT)_{d+1} \cdot R(\pi/2)$. Let us now check that $U$ and $(CPT)_d$ commute. First we compute

$$U \cdot (CPT)_d = T \cdot (CPT)_{d+1} \cdot (CPT)_d = T \cdot (CPT)_{d+1} \cdot R(-\pi/2) \cdot (CPT)_{d+1} \cdot R(\pi/2). \quad (9)$$

Using $R(-\pi/2) \cdot P = P \cdot R(\pi/2)$ and that $T, C$ commute with rotations[3] we get

$$= T \cdot (CPT)_{d+1} \cdot (CPT)_{d+1} \cdot R(\pi) = T \cdot R(\pi), \quad (10)$$

where we have used (8).

On the other hand,

$$(CPT)_d \cdot U = (CPT)_d \cdot T \cdot (CPT)_{d+1} = R(-\pi/2) \cdot (CPT)_{d+1} \cdot R(\pi/2) \cdot T \cdot (CPT)_{d+1}. \quad (11)$$

Using again $R(-\pi/2) \cdot P = P \cdot R(\pi/2)$ we get

$$R(-\pi) \cdot (CPT)_{d+1} \cdot T \cdot (CPT)_{d+1} = (-1)^F R(-\pi) \cdot T, \quad (12)$$

where we used (8) again as well as (7). Since $R(2\pi)$ is the same as $(-1)^F$ we find that (10) and (12) exactly agree, hence $U$ commutes with $(CPT)_d$.

The computation may be repeated beginning with a unitary symmetry which commutes with $(CPT)_{d+1}$ and finding an anti-unitary symmetry on the wall which commutes with $(CPT)_d$ up to $(-1)^F$.

Finally, because of the relation $(CPT)_d = R(-\pi/2) \cdot (CPT)_{d+1} \cdot R(\pi/2)$, it is easy to see that if $(CPT)_{d+1}^2 = 1$, then so does $(CPT)_d$.

---

[2]To decide between these two options one has to be careful about what is meant by $P$. When we write $P$ we always mean a reflection in one coordinate (and not, for instance, a reflection of 3 coordinates in $3+1$ dimensions as is often used in the literature).

[3]Angular momentum is odd under $T$, thus, by anti-unitarity, rotations are even

## 2.2 The 4-Periodic Hierarchy and Some Generalizations

Now let us see how the 4-periodic dimensional hierarchy for $\mathbb{Z}_2$ symmetries (2) follows from the three properties (6),(7),(8).

First, take $U = T \cdot (CPT)$. $U$ is clearly unitary. Further,

$$
\begin{aligned}
U^2 &= T \cdot (CPT) \cdot T \cdot (CPT) = (\text{using (7)}) = (-1)^F T \cdot T \cdot (CPT) \cdot (CPT) = \\
&= (-1)^F T^2 \cdot (CPT)^2 = (\text{using (8)}) = (-1)^F T^2.
\end{aligned}
\tag{13}
$$

So if $T^2 = (-1)^F$, $U^2 = 1$ and vice versa.

Next, we take $T' = U \cdot CPT$ with $U$ a unitary internal symmetry. $T'$ is anti-unitary, and

$$
\begin{aligned}
T'^2 &= U \cdot (CPT) \cdot U \cdot (CPT) = (\text{using (6)}) = U \cdot U \cdot (CPT) \cdot (CPT) = \\
&= U^2 \cdot (CPT)^2 = (\text{using (8)}) = U^2.
\end{aligned}
\tag{14}
$$

So $T'^2 = U^2$.

We can generalize this as follows. Given any symmetry group $G$ with a homomorphism

$$
\varphi : G \to \mathbb{Z}_2,
$$

we can arrange a spontaneous symmetry breaking pattern involving a single real order parameter transforming by

$$
\phi \mapsto (-1)^{\varphi(g)} \phi,
$$

which breaks $G$ down to the kernel $H$ of $\varphi$. Let $U_g$ be the operator corresponding to $g \in G$ (unitary or anti-unitary). There is a $G$-symmetry on the domain wall of the order parameter generated by

$$
\tilde{U}_g = U_g \cdot (CP_\perp T)^{\varphi(g)}.
\tag{15}
$$

(Note that there may still be nontrivial degrees of freedom in the bulk, e.g. if the unbroken symmetries in the kernel of $\varphi$ are anomalous. In any case, $\tilde{U}_g$ is a symmetry, but may act on both bulk and localized degrees of freedom, depending on the situation.)

Another interesting class of examples are theories with a time reversal symmetry $T$ which squares to a unitary $\mathbb{Z}_2$ symmetry $T^2 = U$, with $U^2 = 1$. Such $\mathbb{Z}_4$ time reversal symmetry transformations appear, for instance, in gauge theories in $2+1$ dimensions where $U$ arises from a mod 2 magnetic symmetry [48, 49, 53]. We can imagine breaking $T$ spontaneously with $U$ unbroken, corresponding to the map $\mathbb{Z}_4 \to \mathbb{Z}_2$. Repeating the computations above, we find that on the wall we have a unitary symmetry $V$, $V = T \cdot CP_\perp T$, such that

$$
V^2 = U \cdot (-1)^F.
\tag{16}
$$

Therefore, the theory on the wall now enjoys a unitary $\mathbb{Z}_4$ symmetry. If we performed this procedure again, we would obtain an anti-unitary symmetry $T$ with $T^2 = U \cdot (-1)^F$. It looks like we obtain 4-periodicity, but note that this group, as an extension of $\mathbb{Z}_4$ by fermion parity, actually splits: by the innocuous redefinition $U \mapsto U \cdot (-1)^F$ it becomes the same algebra we started with. Thus we actually obtain a 2-periodic hierarchy unlike in (2). Note that once we make the redefinition $U \mapsto U \cdot (-1)^F$ the original symmetry algebra now takes the form $T^2 = U(-1)^F$. Hence, once one repeats the domain wall construction twice this factor of $(-1)^F$ is physical and cannot be removed. But the hierarchy is still 2-periodic since the symmetry groups $T^2 = U(-1)^F$ and $T^2 = U$ are isomorphic as symmetry groups.

It is also possible to consider symmetry breaking patterns with multiple order parameters, forming a linear representation $V$ of $G$. For instance, we can break $\mathbb{Z}_4$ down to nothing with two real order parameters transforming in the $\pi/2$-rotation representation of $\mathbb{Z}_4$. Such a

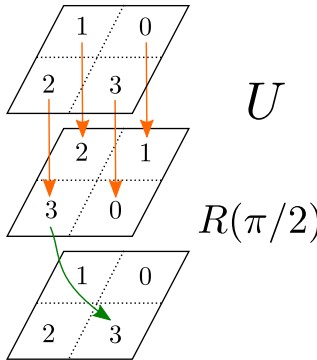

Figure 2: With two real order parameters $\phi_{1,2}$ fully breaking a $\mathbb{Z}_4$ symmetry we have four ground states, labelled $0, 1, 2, 3$. These can be identified with the four signs of the VEVs of the two order parameters $\phi_{1,2} = \pm$. Choosing a boundary condition for $\phi_{1,2}$ such that they wind the unit circle at infinity along a pair of spatial coordinates we obtain a codimension-2 junction where four domain walls coalesce. The global $\mathbb{Z}_4$ symmetry does not act on this junction but we can combine it with a $\pi/2$ rotation to obtain a $\mathbb{Z}_4$ symmetry of the junction. In this case, we did not use $CPT$, so both $\mathbb{Z}_4$ symmetries are unitary.

theory has four vacua $|\text{VAC}_k\rangle$, $k = 0 \ldots 3$, related by the action of the generator of $\mathbb{Z}_4$, $U$, by $U|\text{VAC}_k\rangle = |\text{VAC}_{k+1}\rangle$, with $k$ defined mod 4.

In this situation we have a domain wall between any pair of vacua, but there will be no way to assign symmetries to them. However we can consider a 4-way junction of domain walls, where each of the four vacua meet at a corner. Let us suppose they are ordered $0, 1, 2, 3$ counterclockwise over the four quadrants of the plane. Then while $U$ is not a symmetry of the boundary conditions, $U \cdot R(\pi/2)$ is, where $R(\pi/2)$ is a $\pi/2$ counterclockwise rotation. We see $U \cdot R(\pi/2)$ is unitary and satisfies $(U \cdot R(\pi/2))^4 = (-1)^F$. If it is possible to trivially gap out the domain wall degrees of freedom, we can consider this as a symmetry of the codimension-2 system at the 4-way junction.

If we further arrange for such a symmetry to be spontaneously broken to $(-1)^F$ we can repeat the procedure and find that at the junction inside the junction there is an ordinary $\mathbb{Z}_4$ symmetry. Similar arguments apply in the case that the original $\mathbb{Z}_4$ is anti-unitary.

Thus we find two 2-periodic structures

$$\left\{\begin{matrix}\text{unitary}\\U^4=1\end{matrix}\right\} \implies \left\{\begin{matrix}\text{unitary}\\U^4=(-1)^F\end{matrix}\right\} \implies \left\{\begin{matrix}\text{unitary}\\U^4=1\end{matrix}\right\}, \tag{17}$$

$$\left\{\begin{matrix}\text{anti}-\text{unitary}\\T^4=1\end{matrix}\right\} \implies \left\{\begin{matrix}\text{anti}-\text{unitary}\\T^4=(-1)^F\end{matrix}\right\} \implies \left\{\begin{matrix}\text{anti}-\text{unitary}\\T^4=1\end{matrix}\right\}. \tag{18}$$

In Section 4 we will show how to derive the hierarchies for general symmetry groups and symmetry-breaking patterns.

## 2.3 Anomalies

It is now time to explain how the anomaly of the induced symmetry on the wall relates to the anomaly of the broken symmetry. The discussion in this subsection is a little technical and the reader interested in seeing some explicit examples demonstrating the general results can skip directly to Section 3. (It is in Section 3 that we present the concrete anomaly matching rules for the $\mathbb{Z}_2$ class.)

We will work in the context of anomaly-inflow. Let $Z_{\text{SPT}}(X,A)$ be the partition function of the SPT on a $D+1$ dimensional spacetime $X$, possibly with boundary, equipped with a background gauge field $A$ in a fixed gauge, and let $Z_{\text{dyn}}(Y,A')$ likewise be the partition function of our anomalous theory on a $D$ dimensional closed spacetime with background $A'$, also in a fixed gauge. Anomaly in-flow is the condition that the following combination is gauge invariant (independent of the gauge):

$$Z_{\text{SPT}}(X,A)Z_{\text{dyn}}(\partial X,A|_{\partial X}). \tag{19}$$

This allows us to characterize the anomaly by studying the associated SPT.[4] Moreover, this SPT is determined by its partition functions on closed manifolds, which are topological invariants of $(X,A)$ as well as gauge invariant.

We will see below that when one restricts the gauge field to a submanifold carrying localized degrees of freedom, the type of symmetry that the gauge field couples to can change, depending on the normal bundle of the submanifold. However, because the normal bundle of $\partial X$ is canonically trivial, $A$ and $A|_{\partial X}$ couple to the same kind of $G$ symmetry–that is, an (anti-)unitary element on the boundary acts (anti-)unitarily in the bulk, and an element squaring to $(-1)^F$ on the boundary squares to $(-1)^F$ in the bulk.

Suppose for now $G = \mathbb{Z}_2$. We can break the symmetry simultaneously in the bulk and the boundary, such that a bulk domain wall terminates on a boundary domain wall. If we suppose also that the SPT has no gravitational component, then

$$Z_{\text{SPT}}(X,A) = Z_{\text{SPT}'}(Y,A|_Y), \tag{20}$$

where $Y$ is the worldvolume of the bulk domain wall and $Z_{\text{SPT}'}$ is a $D$ dimensional SPT partition function we interpret in a moment. We assume that in the case of no gravitational anomaly the boundary theory may be trivially gapped away from the domain wall, so that we can continuously deform

$$Z_{\text{dyn}}(\partial X,A|_{\partial X}) \to Z_{\text{dyn}'}(\partial Y,A|_{\partial Y}), \tag{21}$$

where the right hand side is the partition function of the degrees of freedom localized on the boundary domain wall[5]. It follows that up to adding some boundary-local counterterms,

$$Z_{\text{SPT}'}(Y,A|_Y)Z_{\text{dyn}'}(\partial Y,A|_{\partial Y}) \tag{22}$$

is gauge invariant. Thus we obtain an anomaly matching condition between our theory and the modes on the wall, in the case of vanishing gravitational anomaly.

Indeed, (20) says that we can obtain the bulk SPT, hence the anomaly of $Z_{\text{dyn}}$ just knowing $Z_{\text{SPT}'}(Y,A|_Y)$, which (22) says is captured by the anomaly of the domain wall. Note however that (20) only captures $Z_{\text{SPT}'}$ for spacetimes and gauge backgrounds which appear as a domain wall. In Section 4 we will explore this class. A conclusion there is that these spacetimes and gauge backgrounds do not completely capture $Z_{\text{SPT}'}$, which means that the anomaly of the domain wall can be ambiguous. We will see an example of this in Section 3.1.

The restriction $A|_Y$ requires some discussion. We would like to interpret it as a gauge field of a symmetry action on the domain wall degrees of freedom along $Y$. First, observe that the domain wall need not be orientable even though $X$ is. An example is $X = \mathbb{RP}^3$ with its non-trivial $\mathbb{Z}_2$ gauge field being Poincaré dual to an embedded $\mathbb{RP}^2$. In fact, we have the identification

$$w_1(NY) = A|_Y, \tag{23}$$

---

[4]So far, all known anomalous theories can be paired with an SPT satisfying anomaly in-flow, although there can be difficulties in identfying the proper symmetry algebra [30].

[5]If we cannot nondegenerately gap out the system away from the domain wall, e.g. if there is a nontrivial TQFT leftover on either side of it, then we do not have a simple characterization of the domain wall anomaly.

where $NY$ is the normal bundle of $Y$ and $w_1$ is the first Stiefel-Whitney class, which measures the obstruction to choosing a section of $NY$. Indeed, $A|_Y$ may be identified with the self-intersection of $Y$, or the zero set of a generic section of $NY$, which gives the above identification.

Meanwhile, the tangent bundle of $X$ splits along $Y$ as

$$TX|_Y = NY \oplus TY, \tag{24}$$

and since $TX$ is oriented, we get an identification

$$w_1(NY) = w_1(TY) = A|_Y. \tag{25}$$

This means that the symmetry on the domain wall which couples to $A|_Y$ simultaneously reverses the orientation of $TY$ and $NY$. This is equivalent (up to rotations in $TY$) to a $\pi$-rotation in a plane containing $NY$, which agrees with our identification of the symmetry of the domain wall as a $CPT$ transformation (which in Euclidean signature is a $\pi$-rotation) combined with our internal symmetry.

Thus, we interpret the right hand side of (20) as the partition function of a $D$ dimensional SPT obtained on the bulk domain wall, protected by a symmetry which combines the original internal symmetry and the $CPT$ action with $P$ reflecting the normal coordiante.

It is clear that this realization of $CPT$ (ie. a $\pi$-rotation) commutes with all internal symmetries. We will discuss also its relationship with the fermion parity and its commutation relation with anti-unitary symmetries from this point of view in Section 4.

For groups other than $\mathbb{Z}_2$, there are in general several different ways of breaking the symmetry, partially or totally. For instance, if $G$ has a map to $\mathbb{Z}_2$, or equivalently a one dimensional real representation, then we can construct a $G$-symmetric domain wall with action (15). In order to trivially gap the degrees of freedom away from the domain wall we need there to be both no gravitational anomaly and no 't Hooft anomaly when the symmetry is restricted to the unbroken subgroup. In such a case, we can dimensionally reduce the anomaly calculation as above.

More interesting is the case $\mathbb{Z}_n$ with $n$ odd. This group has a single nontrivial irreducible representation (up to automorphisms), given by the $2\pi/n$ rotation of $\mathbb{R}^2$, which represents the target space of two real order parameters, like we have discussed above for $\mathbb{Z}_4$. There is a codimension-2 defect associated with the symmetry breaking pattern where the $n$ vacua meet at a corner. Let $Y \subset X$ be the $D-1$ dimensional worldvolume associated to this defect inside the $D+1$ dimensional spacetime of the SPT in the anomaly in-flow setup. To have a relation like (20) we need to assume both that there is no gravitational anomaly and that the $\mathbb{Z}_n$ domain wall (which doesn't have any internal symmetries) also has no gravitational anomaly. Likewise in this case we expect to be able to trivially gap the boundary theory away from the boundary defects $\partial Y$, giving an anomaly matching our original theory and the theory on the defect with two fewer dimensions. We will prove this matching in Section 4.5.

## 3 Examples and Matching Rules

### 3.1 A Majorana Fermion in $2+1$ Dimensions

We will be using Lorentzian signature $(-,+,+)$ in $2+1$ dimensions. Our gamma matrices are[6]

$$\gamma^0 = i\sigma^2 \,, \qquad \gamma^1 = \sigma^1 \,, \quad \gamma^2 = \sigma^3 \,. \tag{28}$$

They satisfy

$$\{\gamma^\mu, \gamma^\nu\} = 2\eta^{\mu\nu} \,, \qquad [\gamma^\mu, \gamma^\nu] = 2\epsilon^{\mu\nu\rho}\gamma_\rho \,. \tag{29}$$

The Majorana fermion is a real two dimensional spinor $\lambda_\alpha$. The Lagrangian of a massive Majorana fermion is

$$\int d^3x \left( i\bar\lambda\gamma^\mu\partial_\mu\lambda + iM\bar\lambda\lambda \right) \,. \tag{30}$$

Time reversal symmetry and parity act as follows:

$$T: \lambda(x^0, x^1, x^2) \to \pm\gamma^0\lambda(-x^0, x^1, x^2) \,, \tag{31}$$

$$P: \lambda(x^0, x^1, x^2) \to \pm\gamma^1\lambda(x^0, -x^1, x^2) \,. \tag{32}$$

The signs are uncorrelated and arbitrary in principle. We can change the signs at will by combining $P, T$ with fermion number symmetry $(-1)^F$.

The mass term satisfies $T(\bar\lambda\lambda) = -\lambda^T\gamma^0\gamma^0\gamma^0\lambda = \bar\lambda\lambda$. Taking into account the factor of $i$ in the mass term (which is necessary to get a real action) and that $T$ is anti-linear, we find that the under time reversal symmetry $M \to -M$. The same is true under parity. By contrast, the kinetic term is both time reversal and parity invariant.

The theory does not have a notion of charge conjugation symmetry—there are no unitary non-space time symmetries other than $(-1)^F$. Therefore $CPT$ in this theory is just $PT$, modulo the sign choice of how it acts on the fermion, corresponding to including a $(-1)^F$ in the definition. Three important properties that we can readily verify are

$$\begin{aligned} (PT)^2 &= 1 \,, \\ T^2 &= (-1)^F \,, \\ T \cdot PT &= (-)^F PT \cdot T \,. \end{aligned} \tag{33}$$

These properties agree with (6), (7),(8), and the three properties are independent of whether or not one inserts additional factors of $(-1)^F$ in the definitions of $P$ and/or $T$.

There is a remarkable anomaly of the massless Majorana. Since with $M = 0$ the theory has time reversal symmetry, it is in principle possible to ask about gauging it. This means that we could study the massless Majorana fermion on unorientable manifolds with Pin$^+$ structure (since $T^2 = (-1)^F$ [1]). It turns out that there is an obstruction to doing so which is valued in $\mathbb{Z}_{16}$. In other words, if we had 16 Majorana fermions and we defined time reversal symmetry

---

[6]We take the sigma matrices to be

$$\sigma^1 = \begin{pmatrix} 0 & 1 \\ 1 & 0 \end{pmatrix}, \quad \sigma^2 = \begin{pmatrix} 0 & -i \\ i & 0 \end{pmatrix}, \quad \sigma^3 = \begin{pmatrix} 1 & 0 \\ 0 & -1 \end{pmatrix}. \tag{26}$$

They satisfy the usual relations

$$\{\sigma^i, \sigma^j\} = 2\delta^{ij} \,, \qquad [\sigma^i, \sigma^j] = 2i\epsilon^{ijk}\sigma^k \,, \tag{27}$$

where $\epsilon^{123} = 1$.

to act on all of them with the same sign in (31), then we could consistently place the system on unorientable $\text{Pin}^+$ manifolds. This obstruction is interpreted as a $\mathbb{Z}_{16}$ 't Hooft anomaly.

More generally, theories with time reversal symmetry and $T^2 = (-1)^F$ have such an anomaly $\nu_3$ valued in $\mathbb{Z}_{16}$ [34,53,54] (see also references therein). For theories of free fermions, if our time reversal symmetry acts on $N_+$ Majorana fermions with sign $+$ in (31) and sign $-$ on $N_-$ fermions then the time reversal anomaly is given by

$$\nu_3 = N_+ - N_- \quad \text{mod } 16 . \tag{34}$$

Let us see how to derive this interesting fact from domain wall constructions. To that end we would like to break the time reversal symmetry spontaneously, but this is very hard to arrange in a controlled fashion in a theory of Majorana fermions only. However we can modify the theory without changing the anomaly and achieve a simple setup where time reversal symmetry breaking occurs.

We add to the Majorana fermion a real pseudo-scalar $\phi$ coupled via

$$\mathcal{L} = \mathcal{L}_{\text{Kinetic}} + i\phi \bar{\lambda}\lambda + V(\phi^2) . \tag{35}$$

$\mathcal{L}_{\text{Kinetic}}$ includes the usual kinetic terms for the pseudo-scalar and Majorana fermion. $V(\phi^2)$ is an arbitrary potential.

This theory has no non-space time symmetries (other than the usual $(-1)^F$). However it has time reversal symmetry which acts on the fermion as before (31) and on the pseudo-scalar as

$$T : \phi(x^0, x^1, x^2) \rightarrow -\phi(-x^0, x^1, x^2) . \tag{36}$$

It is clear that the time reversal anomaly of this theory is the same as that of the massless Majorana fermion. This is easiest to see by turning off the Yukawa interaction $i\phi\bar{\lambda}\lambda$ and then giving $\phi$ a large mass. We can then integrate out $\phi$ and arrive back at the free Majorana fermion theory.

But now we can also arrange the potential $V(\phi^2)$ such that $\phi$ condenses, for instance by taking $V(\phi^2) = m_\phi^2 \phi^2 + \phi^4$, with $m_\phi^2$ large and negative. This leads to two minima $\phi = \pm v$ and time reversal symmetry is spontaneously broken (parity is broken too, but $PT$ is not spontaneously broken). Note that these minima are both gapped since the fermion acquires an effective mass due to the VEV of $\phi$.

We can now require that for $x^2 \rightarrow \infty$ we approach the vacuum $\phi(\infty) = v$ and for $x^2 \rightarrow -\infty$ we approach the vacuum $\phi(-\infty) = -v$. The system then autonomously finds the least energy configuration with these prescribed boundary conditions at infinity. This configuration is the domain wall, and at low energies it looks like a $1+1$ dimensional object. The domain wall is not invariant under time reversal (36) since it clearly breaks the boundary conditions on $\phi$. However, as explained above, consider the transformation $T \cdot (CP_\perp T) = TP_2 T$. It acts on the pseudoscalar $\phi$ and the fermion $\lambda$ by (taking the plus signs in (31),(32))

$$\phi(x^0, x^1, x^2) \rightarrow -\phi(x^0, x^1, -x^2) ,$$
$$\lambda(x^0, x^1, x^2) \rightarrow \gamma^0 \gamma^2 \gamma^0 \lambda(x^0, x^1, -x^2) = \gamma^2 \lambda(x^0, x^1, -x^2) . \tag{37}$$

The transformation of the pseudoscalar $\phi$ now clearly leaves the domain wall configuration invariant (in this particular model $TP_2 T = P_2$).[7]

While in many familiar examples, such as the Ising model, the domain wall would not support any light degrees of freedom other than the translational zero mode, here the theory

---

[7]More precisely, it leaves the domain wall invariant if the center of mass is at the origin. The same comment applies to the general construction. We can always further combine our transformation with a translation so that the general domain wall configuration is left invariant.

on the wall is richer [55]. Since the bulk is gapped, at low energies we therefore have a genuine $1 + 1$ dimensional theory on the wall. We can identify the light degrees of freedom on the domain wall by solving the equations of motion of the fermion in the presence of the domain wall

$$\gamma^\mu \partial_\mu \lambda = \phi(x^2)\lambda \,, \tag{38}$$

which we treat by separation of variables, $\lambda = h(x^2)\tilde{\lambda}(x^0, x^1)$, such that

$$(\gamma^0 \partial_0 + \gamma^1 \partial_1)\tilde{\lambda} = 0 \tag{39}$$

and

$$\gamma^2 \partial_2 h(x^2)\tilde{\lambda} = \phi(x^2)h(x^2)\tilde{\lambda} \,. \tag{40}$$

The point is that (39) has two normalizable solutions which are left and right moving corresponding to $\gamma^2 \tilde{\lambda} = \pm \tilde{\lambda}$. Therefore for positive $v$ we have to take $\gamma_2 \tilde{\lambda} = -\tilde{\lambda}$ and thus

$$v > 0: \ \gamma_2 \tilde{\lambda} = -\tilde{\lambda} \,, \quad h(x^2) = e^{-\int dx^2 \phi(x^2)} \,, \tag{41}$$

and similarly

$$v < 0: \ \gamma_2 \tilde{\lambda} = +\tilde{\lambda} \,, \quad h(x^2) = e^{+\int dx^2 \phi(x^2)} \,. \tag{42}$$

The fermion is clearly localized to the wall as its wave function $h(x^2)$ decays exponentially far from the wall. The theory on the wall therefore has a chiral fermion, whose chirality is determined by whether $v$ is positive or negative (or the sign of the Yukawa coupling).

Under the residual symmetry on the wall, $TP_2T$,

$$TP_2T : \lambda \to \gamma_2 \lambda = -h(-x^2)\text{sgn}(v)\tilde{\lambda} = -\text{sgn}(v)\lambda \,.$$

As a result, the fermion that originally transforms under time reversal symmetry as $\lambda \to \gamma^0 \lambda$, when it is stuck to the wall, picks up a minus sign if it is right moving and a plus sign if it is left moving. This is an ordinary $\mathbb{Z}_2$ symmetry which we can denote by $(-1)^{F_R}$. In short, one could say that the original time reversal symmetry becomes $(-1)^{F_R}$, which is an ordinary $\mathbb{Z}_2$ symmetry.

Unitary $\mathbb{Z}_2$ symmetries of $1 + 1$ dimensional theories have a $\mathbb{Z}_8$ 't Hooft anomaly (in the bosonic case, where the anomaly is $\mathbb{Z}_2$, the anomaly is connected with the charge of the twisted sector as shown in detail in [56])[8]. More concretely, for theories of free fermions, if we have some number of fermions charged under $(-1)^{F_R}$, then the associated 't Hooft anomaly for $(-1)^{F_R}$ is the number of such fermions mod 8.[9] There is no way for this anomaly to determine the $v_3 \in \mathbb{Z}_{16}$ anomaly of the $2 + 1$ dimensions theory. However, if we combine this $\mathbb{Z}_8$ invariant $v_2$ with the gravitational anomaly of the domain wall theory, that is $2(c_R - c_L) \in \mathbb{Z}$, where $c_R$ and $c_L$ are the central charges of the right-moving and left-moving sectors,[10] respectively, then we find

$$v_3 = 2v_2 - 2(c_R - c_L) \mod 16. \tag{43}$$

As we argue in Section 4.4, there must be a linear relationship between these three quantities, so to verify the anomaly relation (43) we only need to check a couple of cases.

First of all, for the $N_+ = 1$, $N_- = 0$ theory discussed above with $v > 0$, the domain wall carries a right-moving fermion $c_R - c_L = 1/2$ which is odd under $TP_2T$, yielding $v_2 = 1$, from which we find $v_3 = 1$ from (43), matching (34). On the other hand for $v < 0$ the domain wall

---

[8]In non-chiral theories the $\mathbb{Z}_2$ anomaly $v_2$ manifests itself as the spin of the ground states in the twisted sector. For chiral theories, to obtain $v_2$ we have to look at the difference in spin between the twisted and untwisted sectors.

[9]This is closely related to the subject of GSO projection in string theory, see [5, 58, 59].

[10]For local theories of fermions, $2(c_R - c_L)$ must be an integer by the quantization of gravitational Chern-Simons terms.

carries a left-moving fermion $c_R - c_L = -1/2$ which is even under $TP_2T$, yielding $\nu_2 = 0$, from which we again find $\nu_3 = 1$, matching (34). These two examples determine (43) uniquely.

These two examples illustrate that as we change the coupling constants of a theory, there may be multiple domain walls with different anomalies, but they will all have to satisfy the relation (43). Another interesting set of examples can be constructed in the $N_+ = 2, N_- = 0$ theory (two copies of the $N_+ = 1$ theory) because now we can choose the signs of the Yukawa couplings independently between the two fermions. The three cases are:

$$++: \qquad c_R - c_L = 1, \quad \nu_2 = 2 \tag{44}$$

$$+-: \qquad c_R - c_L = 0, \quad \nu_2 = 1 \tag{45}$$

$$--: \qquad c_R - c_L = -1, \quad \nu_2 = 0. \tag{46}$$

We see all three of them match $\nu_3 = 2$ by (43). Note that a nonchiral domain wall with $c_R = c_L$ is only possible if $\nu_3 \in 2\mathbb{Z}$ by (43)!

## 3.2 The $\mathbb{CP}^1$ Model in $1+1$ Dimensions

The main purpose of our next example is to derive an analog of (43) when reducing from $1+1$ dimensions to quantum mechanics in a bosonic system with a unitary $\mathbb{Z}_2$ symmetry. To gradually warm up to the main example, let us first start with free $U(1)$ gauge theory in $1+1$ dimensions,

$$\mathcal{L} = -\frac{1}{2e^2}F^2 + \frac{\theta}{2\pi}F , \tag{47}$$

where $F = da$ is the field strength of the dynamical $U(1)$ gauge field $a$. Charge conjugation symmetry acts by $a_\mu \to -a_\mu$. Time reversal symmetry acts by

$$T: a_x(x,t) \mapsto a_x(x,-t),$$

$$a_t(x,t) \mapsto -a_t(x,-t),$$

and parity acts by

$$P: a_x(x,t) \mapsto -a_x(-x,t),$$

$$a_t(x,t) \mapsto a_t(-x,t).$$

In these conventions, the term $\frac{\theta}{2\pi}F$ breaks time reversal, parity, and charge conjugation symmetry for generic $\theta$. The unbreakable combination, hence our canonical $CPT$, is $PT$. Under $PT$ the term $\frac{\theta}{2\pi}F$ is invariant. A very important fact is that while $\frac{\theta}{2\pi}F$ is odd under time reversal, parity, and charge conjugation, because $\frac{1}{2\pi}F$ has integer integrals over closed two-cycles, all three discrete symmetries are preserved at $\theta = \pi$ (and also obviously at $\theta = 0$). The theory with generic $\theta$ is only invariant under $PT$, $CP$, and $CT$.

The theory at generic $\theta$ has one ground state. For $0 \le \theta < \pi$ the expectation value of the electric field in the vacuum is $\langle F_{tx} \rangle = \frac{e^2 \theta}{2\pi}$. The symmetries $PT$, $CP$, and $CT$ are all unbroken, consistently with the vacuum being unique.

At $\theta = \pi$ a first order transition occurs and another degenerate vacuum appears where the expectation value of the electric field is $\langle F_{tx} \rangle = -\frac{e^2}{2}$. The symmetries $PT$, $CP$, and $CT$ are all unbroken but now $C,P,T$ are all *spontaneously* broken.

It is interesting to ask about the domain wall between these two vacua at $\theta = \pi$. However, the equations of motion of the theory (47) force the electric field to be constant in space and hence a domain wall between these two vacua would have infinite tension. So the example of

pure $U(1)$ gauge theory in $1+1$ dimensions is somewhat exceptional because it has degenerate vacua but the potential barrier between them is infinite.[11]

The absence of a finite-tension domain wall can be interpreted as follows: Since the electric field on one side of the wall is $-\frac{e^2}{2}$ and on the other side $\frac{e^2}{2}$, by the Gauss law, we can interpolate between the two vacua with the aid of a charged particle of charge 1. But since the pure gauge theory does not have dynamical charged particles, the tension (which is the worldline mass of the particle) seems infinite. Therefore the Wilson line of a charge 1 particle serves as a wall between the two vacua but it is not dynamical.

In order to have dynamical domain walls let us therefore add to the theory a charged scalar $\Phi$:

$$\mathcal{L} = -\frac{1}{2e^2}F^2 + \frac{\theta}{2\pi}F + |D_a\Phi|^2 + V(|\Phi^2|) \,, \tag{48}$$

where $V(\Phi)$ is a gauge invariant potential for the charged scalar and $D_a\Phi = \partial\Phi + ia\Phi$. We will take the scalar $\Phi$ to be massive, i.e. $V(|\Phi^2|) = M^2|\Phi^2| + \lambda|\Phi^4|$ for $M^2$ positive and large compared to $e^2$. $C$, $P$, and $T$ act in the following way on $\Phi$:

$$C : \Phi(x,t) \to \Phi^*(x,t)\,, \quad T : \Phi(x,t) \to \Phi^*(x,-t)\,, \quad P : \Phi(x,t) \to \Phi(-x,t)\,. \tag{49}$$

It is worth giving an intuitive explanation of why we do not perform a complex conjugation of $\Phi$ in the action of $P$. When we reverse the time and the electric field in a motion of a classical particle we do not get a consistent trajectory, unless we in addition change the sign of the charge of the particle. This is why our time reversal symmetry is accompanied by conjugating $\Phi$. But if we reverse the sign of the electric field along with a reflection in space we do get a consistent trajectory without having to reverse the sign of the charge. One can see this from the Lorentz force law $F = q(E + v \times B)$, which must be $T$-even and $P$-odd.

On a more technical level, the coupling of $\Phi$ to the gauge field $a$ takes place through the combination $ia_\mu(\Phi\partial^\mu\Phi^* - \Phi^*\partial^\mu\Phi)$. The difference between the time reversal and parity is that the former leads to another sign due to the factor of $i$ in front and hence we need to perform a complex conjugation.

At $\theta = \pi$ the theory (50) still has two exactly degenerate vacua because integrating out the massive $\Phi$ cannot lead to terms which break the symmetries $C, P, T$. The domain wall (kink) between the two vacua is just our $\Phi$ particle.

This model clearly has no anomalies. The easiest way to see it is that for $M^2 < 0$ (and large) the $\Phi$ field condenses and we have a single trivial vacuum.

The story becomes much more interesting (and relevant to the main subject of this paper) if there are two (or more) species of $\Phi$:

$$\mathcal{L} = -\frac{1}{2e^2}F^2 + \frac{\theta}{2\pi}F + \sum_{k=1,2}|D_a\Phi^k|^2 + V(|(\Phi^1)^2|, |(\Phi^2)^2|) \,. \tag{50}$$

For generic choices of the potential there is still no anomaly since we can have one of the $\Phi$'s be heavy and the other condense and we would thus end up with a trivial vacuum. However, we can require the following version of charge conjugation symmetry:

$$C' : a_\mu \to -a_\mu\,, \quad \Phi^1 \to (\Phi^2)^*\,, \quad \Phi^2 \to -(\Phi^1)^*\,. \tag{51}$$

This symmetry $C'$ precludes $\Phi^1, \Phi^2$ from having different signs for their mass squared so we cannot drive the system to a trivial phase quite as easily. A subtle point is to note that on the

---

[11]Another way to say it is that on a circle the two vacua at $\theta = \pi$ do not "mix" and remain exactly degenerate. This can be understood due to an anomaly involving one-form symmetry in $1+1$ dimensions which becomes an ordinary symmetry in quantum mechanics. Such anomalies are not limited to $U(1)$ gauge theories and there are many other examples with similar consequences.

scalar fields $(C')^2\Phi^{1,2}(C')^{-2} = -\Phi^{1,2}$. Since this minus sign is a gauge transformation, strictly speaking,$(C')^2 = 1$. But in some sense that we will explain below this minus sign "comes to life" on the domain wall because there is de-confinement there.

In bosonic systems in 1+1 dimensions the anomalies for a $\mathbb{Z}_2$ symmetry are classified by $\mathbb{Z}_2$ as well (the anomaly inflow term is just $i\pi \int A^3$ where $A$ is a background $\mathbb{Z}_2$ gauge field). It turns out that $C'$ has such a 't Hooft anomaly. We will see below several derivations of that fact, starting from the domain wall construction which shows that the domain wall furnishes a Kramers doublet (5).

Consider for instance the potential

$$V(|\Phi^1|^2, |\Phi^2|^2) = M^2(|\Phi^1|^2 + |\Phi^2|^2) + \lambda(|\Phi^1|^4 + |\Phi^2|^4) . \tag{52}$$

This potential obeys the $C'$ symmetry and we can imagine adding various other interactions to the Lagrangian to break all the other discrete symmetries (except for the unbreakable $PT$ symmetry).[12] We take $\theta = \pi$ and $M^2 > 0$. The theory has two vacua, where the degeneracy is protected by $C'$. Unlike in the pure gauge theory, now the domain wall has finite energy since we have the $\Phi$ particles which have charge 1.

As always, to understand what remains of $C'$ on the domain wall we have to compose it with the unbreakable $PT$ symmetry. It is straighforward to see that $C'PT$ acts as follows on the $\Phi$ particles:

$$C'PT : \Phi^1 \to \Phi^2 , \quad \Phi^2 \to -\Phi^1 .$$

This is an anti-unitary symmetry that acts on the domain wall. Therefore, we have established that in this bosonic system the domain wall has two states, corresponding to the excitations $\Phi^{1,2}$, which realize a time reversal symmetry $T' = C'PT$ action as

$$T'|\Phi^1\rangle = |\Phi^2\rangle , \quad T'|\Phi^2\rangle = -|\Phi^1\rangle .$$

In particular, it is impossible to lift the degeneracy on the wall due to this $T'$ symmetry, which satisfies $T'^2 = -1$ and hence realized projectively on the domain wall. This is our Kramers doublet, which has anomaly polynomial $i\pi \int_{\mathcal{M}_2} w_1^2$ [18], where $w_1$ is the orientation class, which can be thought of as the gauge field that couples to $T'$. When we apply our general anomaly-matching of Section 4, we will see this matches the 1+1D anomaly $i\pi \int_{\mathcal{M}_3} A^3$ for the background gauge field that couples to $C'$, as desired.

Let us now give an independent derivation for this anomaly using the results of [29]. Considering the class of models (50), we can choose the potential to preserve an $SU(2)$ global symmetry if the potential is only a function of $|\Phi^1|^2 + |\Phi^2|^2$. The model then automatically also admits the charge conjugation symmetry $C$ (and $C'$). Altogether, taking into account the gauge transformations, the symmetry of the model is $O(3)$. In [29] it was argued that there is an anomaly $i\pi \int_{\mathcal{M}_3} w_3(O(3))$. Restricting to the $\mathbb{Z}_2$ subgroup of the scalar matrix $-1 \in O(3)$, which is our $C'$, we find $w_3(O(3)) = A^3$, so this result is already enough to imply the $C'$ anomaly.

However, there is another possible anomaly for the $O(3)$ symmetry: $i\pi \int_{\mathcal{M}_3} w_1^3(O(3))$, which would also contribute $i\pi \int_{\mathcal{M}_3} A^3$ and lead to a trivial anomaly. We need to show that the $w_1^3$ anomaly is absent for the $O(3)$ symmetry above. To do so, assume towards a contradiction that there were such an anomaly and consider restricting $O(3)$ to the subgroup of the diagonal matrix with eigenvalues $+1, +1, -1$, which is conjugate to $C$. Then we would find an

---

[12]For instance, we can break the "naive" charge conjugation symmetry $C$

$$C : a_\mu \to -a_\mu , \quad \Phi^1 \to (\Phi^1)^* , \quad \Phi^2 \to (\Phi^2)^* $$

by adding the operator $i(\Phi^1(\Phi^2)^* - \Phi^2(\Phi^1)^*)$ with a small coefficient.

anomaly $i\pi \int_{\mathcal{M}_3} A^3$ for $C$. However, we have shown $C$ is anomaly-free by giving a symmetric deformation to a trivial theory, a contradiction, so the total $O(3)$ anomaly is just the $w_3$ term.

Finally, we make some comments on emergent symmery in the model (50) with $\theta = \pi$. If we choose the potential to respect $SU(2)$ symmetry, e.g.

$$V = M^2(|\Phi^1|^2 + |\Phi^2|^2) + \lambda(|\Phi^1|^2 + |\Phi^2|^2)^2, \tag{53}$$

and take $M^2 > 0$ then we have a domain wall with a doubly degenerate ground state, which we have analyzed above in detail. But if we take $M^2 < 0$ the scalar fields condense and we obtain the $\mathbb{CP}^1$ model at $\theta = \pi$. This flows to a conformal field theory which is the $SU(2)_1$ WZW model (see [65] for the references on this classic result). The symmetry of the infrared model is $SO(4)$ in which the ultraviolet $O(3)$ symmetry is contained. The above discussion fixes the anomaly of the $O(3)$ subgroup of $SO(4)$. In fact this is enough to fix the whole $SO(4)$ anomaly. This will be useful in the next subsection.

Indeed, $SO(4)$ has two Chern-Simons levels, one coming from the Pontryagin class and the other from the Euler class [17]. When restricting to $O(3)$ we find that the former yields the $w_1^3$ anomaly while the latter yields the $w_3$ anomaly we want. This follows from the fact that the Euler class of a rank 4 bundle is $w_4$ mod 2, and when we restrict the vector representation of $SO(4)$ to $O(3)$ it splits as the sign of $O(3)$ plus the vector of $O(3)$, and so $w_4$ splits as $w_4(SO(4)) = w_1(O(3))w_3(O(3)) = Sq^1 w_3(O(3))$. Thus the $SO(4)$ anomaly is the level 1 Chern-Simons term corresponding to the Euler class in $H^4(BSO(4), \mathbb{Z})$. We will give another interpretation of this anomaly in the next subsection.

For other anomalies in the 1+1D abelian Higgs model, especially with regards to the anomaly matching between charge conjugation, parity, and time reversal, coming from the $CPT$ theorem, see [62] and references therein. A paper which recently showed the anomaly for the $SU(2)_1$ point is in some sense extremal for $c = 1$ is [9]. For recent generalizations to the $\mathbb{CP}^N$ model, see [8].

## 3.3 The $S^4$ Sigma Model in $2 + 1$ Dimensions

In this section we present another example of the $\mathbb{Z}_2$ anomaly matching in the chains (2),(3) but this time for time reversal symmetry in 2+1 dimensional *bosonic* theories. As one may guess, the domain wall construction leads to a bosonic theory in 1+1 dimensions with a unitary $\mathbb{Z}_2$ anomaly, exactly of the same type studied in detail in the last subsection. So the bosonic time reversal anomaly in 2+1 dimensions that we study here reduces to the $\pi i \int A^3$ anomaly on the domain wall, and that, in turn, can be further reduced by symmetry breaking on the wall to a Kramers doublet in quantum mechanics.

As a general comment, among the renormalizable quantum field theories in 2+1 dimensions, it is not entirely trivial to write a bosonic theory with a time reversal anomaly. One can do it in a theory with fermions which obey a spin-charge relation, so that only bosonic states are gauge invariant. Such constructions typically lead to rather more complicated phases than those we are interested in. So we will instead study a manifestly bosonic theory in that there are no fermions in the Lagrangian nor does the WZW term require a choice of a spin structure, but it will come at the price of being a non-renormalizable sigma model. Of course, the question of renormalizability is not important for the discussion of anomalies which is why we are allowed to proceed.

We study a (bosonic) non-linear sigma model in $2 + 1$ dimensions with target space $S^4$. We denote the field as a real 5-vector $\vec{n} = (n_1, n_2, n_3, n_4, n_5)$ satsifying the constraint $|\vec{n}|^2 = 1$. We also denote by $\Omega$ the volume 4-form on $S^4$ normalized by $\int_{S^4} \Omega = 1$. We take as our action the (Euclidean) WZW action

$$S = \int_X d^3 x (\partial \vec{n})^2 + 2\pi i \int_Z \hat{n}^* \Omega, \tag{54}$$

where $Z$ is a 4-manifold with $\partial Z = X$ and $\hat{n} : Z \to S^4$ restricts to $\vec{n}$ on the boundary. It is always possible to find such a filling $Z$ and extension $\hat{n}$ because $\Omega_3^{SO}(S^4) = 0$. However, the exponentiated action does not actually depend on the choice of $(Z, \hat{n})$ because the coefficient is $2\pi i$ and the periods of $\hat{n}^*\Omega$ over closed 4-manifolds are integers by our normalization.

Besides the $SO(5)$ rotation symmetry of $\vec{n}$, the kinetic term above also has a unitary antipodal symmetry

$$C : \vec{n} \mapsto -\vec{n}, \tag{55}$$

extending the $SO(5)$ group to $O(5)$, as well as spacetime parity and time reversal symmetries $P$ and $T$ acting trivially on $\vec{n}$. However, the WZW term breaks the symmetry (55) and only has the combined symmetries $PT, CP$, and $CT$.

We are interested in the 't Hooft anomalies of $CT$ and of the enlarged group $SO(5) \times \mathbb{Z}_2^{CT}$. First we would like to understand the pure anomalies of the $CT$ symmetry. For this sake we will construct a $CT$ domain wall. Since for now we will not be interested in the $SO(5)$ symmetry, we will (partially) break it in order to simplify the domain wall construction. A particularly simple potential with $SO(4)$ symmetry on $S^4$ may be defined by the square of the "height map"

$$W(\vec{n}) = -n_5^2, \tag{56}$$

which has a minimum at the south pole $n_5 = -1$ and at the north pole $n_5 = 1$ and is maximal over the equatorial $S^3$. This potential breaks explicitly $SO(5) \times \mathbb{Z}_2^{CT}$ down to its $SO(4) \times \mathbb{Z}_2^{CT}$ subgroup. The $SO(4)$ rotates the first four coordinates $(n_1, n_2, n_3, n_4)$ and $CT$ flips the sign of all $n$'s along with reversing the time coordinate.

As in Section 3.1, we will implement this potential by a coupling to a real scalar $\phi$ with total action

$$S' = S + \int_X d^3x \left( (\partial \phi)^2 + V(\phi^2) + \phi n_5 \right), \tag{57}$$

where $V(\phi^2)$ is a Landau-Ginzburg potential. This coupling preserves the subgroup $SO(4) \times \mathbb{Z}_2^{CT}$ where $CT$ acts on $\phi$ by

$$CT : \phi \mapsto -\phi \tag{58}$$

(accompanied by reflection of the time coordinate). We can give $\phi$ a very large mass in $V(\phi^2)$ and the full $SO(5) \rtimes \mathbb{Z}_2^{CT}$ symmetry will be restored, so the $SO(4) \times \mathbb{Z}_2^{CT}$ anomaly of $S'$ must match the $SO(4) \times \mathbb{Z}_2^{CT}$ anomaly of $S$ for the unbroken subgroup. Since we can do this for any $SO(4) \times \mathbb{Z}_2^{CT}$ subgroup by choosing different axes for our coupling potential, this gives a very strong constraint on the $SO(5) \times \mathbb{Z}_2^{CT}$ anomaly of $S$.

To study the $CT$ domain wall, we choose $V(\phi^2)$ in such a way that $\phi$ condenses and we thus have two vacua, related by $CT$. We then choose the frustrated boundary conditions for $x_2 \to \pm\infty$ such that we approach the vacuum $\phi = \pm v$, respectively. The low energy degrees of freedom for $\vec{n}$ will be paths from $n_5 = 1$ at $x_2 = -\infty$ to $n_5 = -1$ at $x_2 = +\infty$. Such paths are given by great semicircular paths from the north to south pole of $S^4$ and are parametrized by the equatorial $S^3$ where the path crosses. Thus, the low energy degrees of freedom on the wall are described by a $1 + 1$ dimensional NLSM with target $S^3$.

Further, the winding number of the crossing points around the equatorial $S^3$ is the same as the winding number of the paths around the big $S^4$, so the level 1 WZW term becomes the level 1 WZW term on the domain wall, yielding the $SU(2)_1$ WZW theory in the infrared on the domain wall.[13]

---

[13] The 2+1D $S^4$ sigma model also arises in the low energy limit of the Higgs phase of $SU(2)$ gauge theory with two fundamental Higgs fields. The WZW level equals the Chern-Simons level of the $SU(2)$ gauge field [2] and this leads to another way to characterize the domain wall.

We must study the induced unitary symmetry on the wall which comes from the sopntaneously broken anti-unitary $CT$. The role of the canonical $CPT$ symmetry in the original $2+1$ dimensional theory is played by $PT$. We choose our $P = P_2$, reflecting $x_2 \mapsto -x_2$. As a result, both $CT$ and $PT$ reverse the domain-wall profile. Their product, $CP$ thus acts as a unitary symmetry $U$ on the domain-wall degrees of freedom. If we write the $S^3$ degree of freedom on the wall as a 4-vector $\vec{l} = (l_1, l_2, l_3, l_4)$ by

$$l_j(x_0, x_1) = n_j(x_0, x_1, 0), \tag{59}$$

we find

$$U : \vec{l} \mapsto -\vec{l}. \tag{60}$$

Meanwhile, the $SO(4)$ enjoyed by the action (57) acts in the usual way on this 4-vector. Therefore, the transformation $U$ in this particular case is in fact part of $SO(4)$. (We could break the original $SO(5)$ symmetry completely, while $U$ would have survived on the domain wall as long as $CT$ is retained.)

Since the theory on the wall flows at long distances to the $SU(2)_1$ WZW model, we can now use the results of the previous section 3.2. There, $U$ acts as a combined charge conjugation and flavor rotation. See also a discussion in [39]. We found in the previous subsection the anomaly

$$\frac{1}{2}A^3 \in H^3(B\mathbb{Z}_2, U(1)), \tag{61}$$

where $A$ is a background $\mathbb{Z}_2$ gauge field coupled to $U$. As we will describe in Section 4.3 (cf. (95)), this implies that $S$ has the $CT$ anomaly

$$\frac{1}{2}w_1(TX)^4 \in \Omega_O^4, \tag{62}$$

which in some sense is similar to $\nu = 8$ in the $\Omega_{Pin^+}^4 = \mathbb{Z}_{16}$ discussed in subsection 3.1. The connection between the bosonic time reversal anomaly (62) and the $\mathbb{Z}_2$ anomaly on the wall (61) is another instance of anomaly matching in our chain (2).

We now return to the problem of fixing the anomaly of our sigma model with $S^4$ target space including the continuous symmetries. In the $1+1$ dimensional $\mathbb{CP}^1$ model the $SO(3) \times \mathbb{Z}_2 = O(3)$ symmetry is manifest and the whole anomaly polynomial may be written

$$\frac{1}{2}A^3 + \frac{1}{2}Aw_2(SO(3)) \in H^3(BO(3), U(1)). \tag{63}$$

For the $2 + 1$ dimensional theory this implies the anomaly

$$\frac{1}{4}w_1(TX)^4 + \frac{1}{2}w_1(TX)^2 w_2(SO(3)) \in \Omega_{SO}^4(BO(3), \xi), \tag{64}$$

where $\xi$ is the fundamental representation of $O(3)$. The first term is what we discussed above, namely the bosonic time reversal anomaly. The second term is interesting – it represents a mixed anomaly between time reversal symmetry and $SO(3)$. Such mixed anomalies between time reversal symmetry and continuous global symmetries are familiar from theories with fermions, but they can also arise in bosonic models, and the $S^4$ sigma model with a WZW term is a nice example of that.

In fact, we can actually use the $SO(4)$ anomaly we derived for the $SU(2)_1$ theory of the previous subsection to fix the $SO(5) \times \mathbb{Z}_2^{CT}$ anomaly directly. We find by inspection the only possibility is given by the twisted Euler class of the vector representation of $O(5)$:

$$e(W) \in H^5(BO(5), \mathbb{Z}^{CT}) = H^4(BO(5), U(1)^{CT}). \tag{65}$$

We have seen an Euler class appear twice for anomalies of different WZW models, and there is a particularly elegant reason why. The Euler class appears very naturally in the study of domain walls, and we discuss it more and give a definition in Section 4.2. For now, we note that one can derive the anomalies above right from the obstruction theory of the WZW term, as was done for continuous symmetries in [38] but which works in general and will work for all symmetries, including anti-unitary ones. It is based on considering the homotopy quotient of the target space by the symmetry group (see [41] for a review). Indeed, gauging a target-space symmetry is the same as extending the theory to one whose target is the homotopy quotient $X//G$, which sits in a fibration

$$X \to X//G \to BG.$$

Then, the anomaly is the obstruction to extending the WZW class from $X$ to $X//G$. When $X$ is an $n$-sphere and $G$ is a subgroup of $O(n+1)$, this obstruction is always the Euler class, see [37].

### 3.4 Some Properties of the Spin $\mathbb{CP}^1$ Model in $2+1$ Dimensions

In this section we clarify some of the basic properties of the Spin $\mathbb{CP}^1$ Model in $2+1$ Dimensions. We discuss its symmetries, its relation to the $\mathbb{CP}^1$ model at $\theta = \pi$, and some of its anomalies. We also remark on the relation of this model to $\text{QCD}_3$ and perform an interesting new consistency check on some conjectured RG flows.

To warm up, we first consider a theory of $2k$-many 2-component complex fermions in $2+1$ dimensions. These can be considered as $4k$-many 2-component Majorana fermions $\lambda_j$. Following our analysis in Section 3.1, we consider a time reversal action (choosing all positive signs)

$$T : \lambda(x^0, x^1, x^1) \mapsto \gamma^0 \lambda(-x^0, x^1, x^2)$$

and parity

$$P : \lambda(x^0, x^1, x^1) \mapsto \gamma^1 \lambda(x^0, -x^1, x^2).$$

As in Section 3.1, we verify $T^2 = (-1)^F$ with the anomaly $4k$ mod 16. Further, $PT$ satisfies the three properties necessary for the canonical $CPT$ symmetry.

This theory also has an interesting $U(2)$ flavor symmetry for which our fermions form $k$ doublets. One checks that this symmetry commutes with $T$ (and $P$), forming the full group $(SU(2) \times U(1) \times \mathbb{Z}_4^T)/\mathbb{Z}_2^F$, which we write as shorthand $U(2) \times T$. The anomaly can be roughly studied as the combination of $SU(2) \times T$ and $U(1) \times T$ anomalies. The former is the $\mathbb{Z}_4$ subgroup $k$ mod 4 of a $\mathbb{Z}_4 \times \mathbb{Z}_2$ classification and the latter is a $\mathbb{Z}_4$ subgroup $2k$ mod 8 of a $\mathbb{Z}_8$ [33].

To make contact with the non-linear sigma model, we would like to couple our fermions to a unit 3-vector field $\vec{n} \in \{(n^1, n^2, n^3) \in \mathbb{R}^3 \mid (n^1)^2 + (n^2)^2 + (n^3)^2 = 1\}$ by the Yukawa coupling (using Lorentz signature)

$$i n^j \bar{\psi} \tau_j \psi,$$

where $\tau_j$ are a basis of the $su(2)$ flavor algebra. In the presence of this coupling, the fermions become gapped and the theory flows to a non-linear sigma model with target $S^2$ we refer to as the spin $\mathbb{CP}^1$ model, with action

$$\int d^3x (\partial n)^2 + i\pi k \text{Hopf}(n), \tag{66}$$

where, as discussed in [28], when $k$ is odd the extra topological term, known as the Hopf term, introduces nontrivial dependence on a spin structure.

One can of course study the sigma model (66) by itself, without the additional fermions and Yukawa couplings (see [66] for some background). This model is also referred to as the

$\theta = \pi \, \mathbb{CP}^1$ model in 2+1 dimensions.[14] The main consequence of the second term in (66) for odd $k$ is to render the Skyrmion into a fermion (which follows from the Callias index theorem).

We find our discrete symmetries act as

$$P, T : n^j \mapsto -n^j,$$

which indeed commute with the obvious $SO(3)$ rotation symmetry of $n^j$. There is also the Skyrmion number $S$, ie. the winding number of $n$ over a spatial slice, which is a conserved charge which generates a symmetry $U(1)_S$. We see that $T$ takes the Skyrmion to the anti-Skyrmion, hence commutes with the $U(1)_S$ group, which is given by $e^{i\alpha S}$.

In fact we can identify the $SO(3) \times U(1)_S$ group of the spin $\mathbb{CP}^1$ model with the quotient of $U(2)$ by the fermion parity, using the Callias index theorem which says that the Skyrmion binds $k$ complex fermionic zero modes. Thus, including the trivial massive fermion the symmetry group of the spin $\mathbb{CP}^1$ model is also $U(2) \times T$. We would like to match the anomaly with the free fermion theory.

The main reason for us to discuss this model here is that there is an apparent difficulty that the action (66) only sees $k$ mod 2 but the anomaly by the free fermion calculation depends on $k$ mod 4. Let us therefore discuss $k$ even and try to find the discrepancy. This apparent discrepancy is very important – as we will later see the same discrepancy arises when we try to match the anomaly of (66) with QCD$_3$ and the resolution of the discrepancy is going to be the same.

The resolution is to realize that the $k = 0$ model, despite being a bosonic theory (when the transparent massive fermions are ignored) has a mixed anomaly between the time reversal symmetry and the Skyrmion number. This is quite similar to what we have found in the previous section about the $S^4$ bosonic WZW model. This anomaly has been derived and described in [29]. Let $F$ be the field strength of the gauge field that couples to $U(1)_S$.[15] Then the anomaly we are talking about takes the form

$$\frac{1}{2} w_1^2 \frac{F}{2\pi}.$$

In [29] a derivation was given in terms of an Abelian Higgs model that flows to the $k = 0$ model (66). For another way to derive this anomaly, one may compactify along an $S^2$ carrying $2\pi$ flux for the $U(1)_S$ gauge field. Because of the nontrivial extension

$$\mathbb{Z}_2 \to U(2) \to SO(3) \times U(1)_S,$$

we find that the resulting $0+1$ dimensional system has a pair of ground states transforming in the spin-1/2 representation of $SO(3)$. Because our time reversal commutes with this action and this representation is quaternionic, we also have $T^2 = -1$ on these ground states, which is the meaning of the above anomaly.

Most importantly for us, this mixed anomaly means that if we redefine time reversal symmetry by the $\mathbb{Z}_2$ subgroup of $U(1)_S$:

$$T \to T(-1)^S, \tag{67}$$

---

[14]Even though the fermions are all gapped due to the Yukawa couplings, we make a distinction between the model with the fermions and the pure model without the fermions for two reasons: First, the model with the fermions has an additional $U(1)$ symmetry inside $U(2)$ which only acts on the heavy fermions and is not present in the pure NLSM (66). Second, the pure NLSM has a spin-charge relation between Skyrmions and fermions while there is no such spin-charge relation in the theory with heavy fermions. These issues would not be important for us but it is useful to keep them in mind.

[15]Since time reversal flips the Skyrmion number, the background gauge field for $U(1)_S$ is invariant (as a one-form) under time reversal. This is important in the argument below.

we shift the pure $T$ anomaly by $\nu = 8$ mod 16. This may be demonstrated by our domain wall picture as well.[16]

The apparent mismatch between the number of fermions being $4k$ mod 16 (hence obeying a time reversal anomaly for even $k$ which is not divisible by 4) and the theory at even $k$ not having a topological term is resolved by the fact that the theory (66) has two possible notions of time reversal symmetry (67). This is nicely reflected in the domain wall construction of this model.

In summary, we have found that for even $k$ the time reversal anomaly of the model (66) is either 0 or 8 mod 16 while for odd $k$ it is either 4 or 12 mod 16, depending on how the time reversal symmetry acts on the Skyrmions.

Let us now contrast this with some conjectures about the infrared behavior of $QCD_3$. Consider $SU(N)$ gauge theory with vanishing Chern-Simons level and $N_f$ fundamental fermions ($N_f$ must be even for consistency). We can pick a time reversal symmetry that commutes with the $U(N_f)$ flavor symmetry. This would act on the two fermion flavors by $\Psi_i \to \gamma_0 \Psi_i^\dagger$ for $i = 1, .., N_f$. This commutes with $U(N_f)$ even though complex conjugation is involved because time reversal is an anti-unitary symmetry. The time reversal anomaly of this model is therefore given by counting the number of Majorana fermions in the ultraviolet and one finds $2N_f N$ mod 16. For $N_f = 2$ we find $4N$ mod 16. It was conjectured [67] that the model flows at long distances to the sigma model (66) with $k = N$. The ultraviolet anomaly is therefore in precise agreement with our result $4k$ mod 16 for the anomaly of the non-linear sigma model (66). It would be nice to repeat this analysis for arbitrary $N_f$, match the other anomalies, and also identify more clearly the mapping of the symmetries.

# 4 Dimensional Reduction and Cobordisms

In this section we will explore the mathematical description of the dimensional reduction picture in terms of the cobordism classification of SPT phases/'t Hooft anomalies. This will allow us to prove a number of interesting results. In particular we will show that for a $\mathbb{Z}_2$ symmetry, the anomaly is always captured by the domain wall. On the other hand, for $\mathbb{Z}_n$ symmetries, one has to study the gravitational response of the domain wall as well as the $\mathbb{Z}_n$ anomaly on the junction, and then one can reconstruct the anomaly. This way, the anomaly of all finite abelian groups may be computed by dimensional reduction, focusing on one cyclic factor at a time. For mixed anomalies involving a finite abelian group and a Lie group $G$, the anomaly calculation can be dimensionally reduced to a pure $G$ anomaly. We expect that, in most cases, by restricting to different abelian subgroups, we can recover an arbitrary anomaly by dimensional reduction.

## 4.1 Bordism and Cobordism Groups

We are interested in the *($\xi$-twisted) (S-)bordism groups* $\Omega_n^S(W, \xi)$, where $W$ is a space, $\xi \to W$ is a real vector bundle over $W$, and $S \to O$ is a kind of stable structure for real vector bundles, meaning that $S$ is a group and an $S$-structure for a bundle is a lift of its transition maps (valued

---

[16]In more detail, at one step of the domain wall construction, we find that the even $k$ theory reduces to the $1 + 1$ dimensional compact boson on the domain wall, analogous to the discussion in Section 3.3. Using the action of $(PT)T$ on the wall, we find a unitary symmetry $U$ which acts as a $\pi$ rotation of the compact boson. As is well known, the anomaly of such an action is not determined unless we know how $U$ acts on the vortex. A nice property we also used above is that the vortex number on the 1d wall equals the Skyrmion number of whole 2d configuration. Thus we see that the two different time reversal symmetries give rise to unitary symmetries with opposite anomalies, as expected. When $U$ acts trivially on the vortex we have $\nu = 0$ mod 16 and when $U$ acts nontrivially we have $\nu = 8$ mod 16.

in the subgroup $O(r) \subset O = O(\infty)$, where $r$ is the rank of the bundle) to $S$[17]. Usually $S = SO$ (orientation) or $S = \text{Spin}$ (spin structure), but it can also be $\text{Spin}^c$ or involve other internal symmetries.

The group $\Omega_n^S(W, \xi)$ consists of equivalence classes $[X, f, s]$, where $X$ is an $n$-manifold, $f : X \to W$ is a map, and $s$ is an $S$-structre on $TX \oplus f^*\xi$ (sometimes called a $\xi$-twisted $S$-structure). These classes form a group by disjoint union with inverses given by orientation-reversal. We have

$$[X, f, s] = 0, \tag{68}$$

whenever there is an $(n+1)$-manifold $Z$ with $\partial Z = X$ and with extensions of $f$ and $s$. Such a manifold is called a ($\xi$-twisted $S$-)nullbordism. A nullbordism of $[X, f, s] - [X', f', s']$ is called a ($\xi$-twisted $S$-)bordism between them. Usually we supress $f$ and $s$ from the notation.

Meanwhile, the *($\xi$-twisted) (S-)cobordism groups* are defined by Anderson duality [33]. This implies there is a short exact sequence

$$\text{Ext}(\Omega_n^S(W, \xi), \mathbb{Z}) \to \Omega_S^n(W, \xi) \to \text{Hom}(\Omega_{n+1}^S(W, \xi), \mathbb{Z}) \tag{69}$$

analogous to the universal coefficient sequence [35]. This sequence splits, meaning

$$\Omega_S^n(W, \xi) = \text{Ext}(\Omega_n^S(W, \xi), \mathbb{Z}) \oplus \text{Hom}(\Omega_{n+1}^S(W, \xi), \mathbb{Z}), \tag{70}$$

but this splitting is non-canonical, meaning that if we have maps of bordism groups, we cannot necessarily use this splitting to compute the map on the cobordism group. We will see an important example of this below.

It has been argued that elements of $\Omega_S^n(W, \xi)$ may be identified with partition functions of invertible TQFTs for spacetime $n$-manifolds $X$ equipped with a map $f : X \to W$ with $\xi$-twisted $S$-structure [1, 33].

For $W = BG$, $S = SO$ (resp. $Spin$) these classify 't Hooft anomalies of bosonic (resp. fermionic) systems in $n-1$ spacetime dimensions with bosonic symmetry $G$. The first factor in (70) can be regarded as the torsion phases (global anomalies), while the second factor contains generalized gravitational Chern-Simons terms. In these cases, the bordism group only depends on

$$w_1(\xi) \in H^1(BG, \mathbb{Z}_2), \tag{71}$$

which represents a homomorphism $G \to \mathbb{Z}_2$ which picks out the anti-unitary elements, and

$$w_2(\xi) \in H^2(BG, \mathbb{Z}_2), \tag{72}$$

which classifies the extension

$$\mathbb{Z}_2^F \to G_{\text{tot}} \to G, \tag{73}$$

where $\mathbb{Z}_2^F$ is generated by the fermion parity operator. In general, the (co)bordism group is independent of any twist $\xi$ which itself admits $S$-structure.

## 4.2 The Smith Maps and Anomaly Matching

Suppose we have a theory with $(G, \xi)$ symmetry in $D$ spacetime dimensions. Its anomaly is characterized by an element

$$\alpha_D \in \Omega_S^{D+1}(BG, \xi). \tag{74}$$

Let $V$ be an $r$ dimensional representation of $G$. We introduce $r$ many real order parameters transforming according to $V$. There is a codimension-$r$ defect where along an $(r-1)$-sphere linking the defect the order parameters wrap the unit $(r-1)$-sphere in $V$. This defect may be

---

[17]Here stable means that an $S$-structure on a pair of bundle $V_1, V_2$ defines an $S$-structure on $V_1 \oplus V_2$.

endowed with a $G$ symmetry using CPT and rotations as we have discussed. Elsewhere, we assume the system is trivially gapped, so this defect has an anomaly

$$\alpha_{D-r} \in \Omega_S^{D-r+1}(BG, \xi \oplus V), \tag{75}$$

where the modified twist comes about because $G$ acts on the normal bundle of the defect in the representation $V$. We will define a map

$$f_V : \Omega_S^{n-r}(BG, \xi \oplus V) \to \Omega_S^n(BG, \xi) \tag{76}$$

such that if our theory may be trivially gapped away from the $V$-defect, then its anomaly satisfies

$$f_V(\alpha_{D-r}) = \alpha_D, \tag{77}$$

and so the defect captures the anomaly. We believe the converse holds as well, compare Section 2.3. This is the case for $G = \mathbb{Z}_2$ and $V = \sigma$ the one-dimensional sign representation in the absence of gravitational anomaly, which corresponds to the $\mathbb{Z}_2$ domain wall we have discussed, and we will show this fact from the geometric point of view. In other cases the cokernel of $f_V$, which is the obstruction to trivially gapping the theory away from the $V$-defect, also involves 't Hooft anomalies. We will describe how this works for a general finite abelian group in Section 4.5.

We will actually define and work mostly with the dual bordism map

$$\Omega_n^S(BG, \xi \oplus V) \to \Omega_{n-r}^S(BG, \xi), \tag{78}$$

which describes the analogous reduction from $(X, A)$ to $(Y, A|_Y)$ in our physical anomaly-matching relation (20).

Suppose $X$ is a closed $n$-manifold endowed with a $G$ bundle, equivalently a map $A : X \to BG$. Let $V$ be an $r$ dimensional real representation of $G$. We can consider $V$ as a $\mathbb{R}^r$ bundle over $BG$. It has an Euler class,

$$[e(V)] \in H^r(BG, \mathbb{Z}^{\det(V)}), \tag{79}$$

where $\mathbb{Z}^{\det(V)}$ denotes integer coefficients twisted by the determinant line of $V$. The pullback $A^*[e(V)] = [e(A^*V)]$ can actually be represented by a codimension $r$ submanifold in $X$ as follows.

First consider the pullback bundle $\pi : A^*V \to X$. We choose a smooth section of this bundle: $s : X \to A^*V$ such that $\pi \cdot s = id$. Locally, $s$ may be represented as an $r$-tuple of smooth real-valued functions $s = (s_1, \ldots, s_r)$. Thus we can modify $s$ locally near each zero so that zero is a regular value. Let us restrict our attention to sections with this property. Then the zero locus of $s$ is a codimension $r$ submanifold $E(s)$ of $X$.

If there are no zeros of $s$, then $s$ generates a trivial sub-bundle of $V$ and hence $[e(V)] = 0$. Moreover, if the zero locus of $s$ is the boundary of an $(r+1)$-chain, then we can modify $s$ near this $(r+1)$-chain so that it is non-vanishing. Hence by usual obstruction theory arguments $E(s)$ is a Poincaré dual representative of $[e(V)]$.

Furthermore, the bordism class of $E(s)$ only depends on the bundle $A^*V$. Indeed suppose $s'$ is another section of $A^*V$, regular at zero. Then we have a 1-parameter family of sections

$$s(t) = ts' + (1-t)s \tag{80}$$

such that $s(0) = s, s(1) = s'$. We can consider this family to be a section over the extended bundle $A^*V$ on $X \times [0, 1]_t$. We perturb this resulting section to be regular at zero, which we can do without modifying anything near $X \times 0$ or $X \times 1$ since the section is already regular at zero there and regularity is an open condition. It follows that the zero locus of this section is a $(n-r+1)$-manifold with boundary $E(s) \sqcup E(s')$, ie. a bordism between them.

Finally, it is clear that if we have a bordism of $(X,A)$ with $(X',A')$, that the two submanifolds so constructed are also bordant.

Thus we have constructed a map

$$\Omega_n^O(BG) \to \Omega_{n-r}^O(BG). \tag{81}$$

If $X$ has tangent structure we can do even better. Indeed, observe that the normal bundle of $E(s)$ is equivalent to the restriction of $V$. Therefore, if $X$ is equipped with structure on $TX \oplus A^*\xi$, where $\xi$ is some vector bundle over $BG$, then $E(s)$ is equipped with that same structure on the restricted bundle,

$$(TX \oplus A^*\xi)|_{E(s)} = TY \oplus (A|_{E(s)})^*(V \oplus \xi). \tag{82}$$

This is likewise true of the bordism above that we constructed, since it is also a zero locus of a section of $A^*V$, and may be assumed likewise of any bordism of $(X,A)$. Therefore, with such tangent structures we actually obtain a map

$$\Omega_n^S(BG,\xi) \to \Omega_{n-r}^S(BG,\xi \oplus V), \tag{83}$$

where $S$ is some structure, such as an orientation, spin structure, or spin$^c$ structure (it could even be another gauge field). We call this the *Smith map*, because it generalizes the Smith isomorphism described in [36]. See also [50]. Taking duals (in the sense of Anderson duality (69)), we further obtain the desired map

$$f_V : \Omega_S^{n-r}(BG,\xi \oplus V) \to \Omega_S^n(BG,\xi), \tag{84}$$

which we also call the Smith map. Note that this map does not need to split according to (69).

We note that one can analogously define Smith maps for any space $W$ equipped with a rank $n$ vector bundle $V$,

$$\Omega_D^S(W,\xi) \to \Omega_{D-n}^S(W,\xi \oplus V). \tag{85}$$

## 4.3 Some Properties of the Smith Maps

We collect here some elementary facts about the Smith maps, some of which are proven in [36] for unitary representations for untwisted bordism.

Consider the map $\Omega_D^S \to \Omega_D^S(BG,\xi)$ given by endowing an $S$-manifold with the trivial $G$ bundle. The cokernel of this map is called the *reduced bordism group* $\tilde\Omega_D^S(BG,\xi)$. This is dual to the *reduced cobordism group* $\tilde\Omega_S^D(BG,\xi)$ which is the subgroup of cobordism invariants which are trivial if the $G$ bundle is trivial. Clearly the image of all cobordism Smith maps are reduced cobordism invariants and the bordism Smith maps descend to reduced bordism classes. We refer to

$$\tilde\Omega_D^S(BG,\xi) \to \Omega_{D-n}^S(BG,\xi \oplus V) \tag{86}$$

and its dual

$$\Omega_S^{D-n}(BG,\xi \oplus V) \to \tilde\Omega_S^D(BG,\xi) \tag{87}$$

as the *reduced Smith maps*. We note that if $\xi = 0$, then the inclusion of bordism groups above splits, since given an $S$-manifold with a $G$ bundle we can forget the $G$ bundle. It follows

$$\Omega_D^S(BG) = \Omega_D^S \oplus \tilde\Omega_D^S(BG). \tag{88}$$

This lets us interpret the elements of $\tilde\Omega_D^S(BG)$ as those $S$-manifolds with $G$-bundles which are $S$-nullbordant after forgetting the $G$-bundle.

If we have $V = V_1 \oplus V_2$, then we can decompose the Smith map of $V$ into a composition of the Smith maps of $V_1$ and $V_2$. This is because a section of $V$ is a section $s_1$ of $V_1$ plus a section

$s_2$ of $V_2$, so its vanishing locus may be taken to be the intersection of the vanishing loci, or equivalently we take the vanishing locus of $s_1$ and consider $s_2$ as a section of $V_2$ restricted to it and then take the vanishing locus of the restricted section, or vice versa. This shows that all Smith maps commute up to signs $(-1)^{n_1 n_2}$ where $n_j = \dim V_j$ from the intersection count. For Euler classes, this means

$$e(V_1 \oplus V_2) = e(V_1) \cup e(V_2). \tag{89}$$

Furthermore, the sum of all $S$-bordism groups

$$\Omega_*^S = \bigoplus_n \Omega_n^S \tag{90}$$

forms a ring under Cartesian product of manifolds. For any $(G, \xi)$ then, the sum of $G$-equivariant $\xi$-twisted $S$-bordism groups

$$\Omega_*^S(BG, \xi) = \bigoplus_n \Omega_n^S(BG, \xi) \tag{91}$$

forms an $\Omega_*^S$-module. It is easy to see that the Smith maps are module homomorphisms, that is they commute with the action of $\Omega_*^S$.

Group cohomology classes provide cobordism invariants by integration, giving a map for finite groups

$$H^n(BG, U(1)^{\det \xi}) \to \Omega_S^n(BG, \xi) \tag{92}$$

for any $S$, landing in the torsion subgroup of the cobordism group. Using Anderson duality applied to $H_*(BG, \mathbb{Z})$, we get more generally a map

$$H^{n+1}(BG, \mathbb{Z}^{\det \xi}) \to \Omega_S^n(BG, \xi), \tag{93}$$

which may include non-torsion pieces such as Chern-Simons terms for Lie groups $G$. If we have a rank $r$ representation $V$, cup product with the Euler class defines a map

$$H^{n+1}(BG, \mathbb{Z}^{\det \xi}) \to H^{n+r+1}(BG, \mathbb{Z}^{\det(\xi \oplus V)}) \tag{94}$$

$$\omega \mapsto \omega \cup [e(V)], \tag{95}$$

which commutes with the above map to the Smith homomorphism of $V$. That is, in group cohomology terms, the Smith map is just given by cup product with the Euler class. For a related discussion in the context of crystalline symmetries, including a computation of Euler classes for all 2d point groups and 3d axial point groups, see [6].

## 4.4 $\mathbb{Z}_2$ Smith Maps and the 4-Periodic Hierarchy

Let us consider the case $G = \mathbb{Z}_2$, which relates directly to our anomaly discussions above. All representations of $\mathbb{Z}_2$ are sums of the trivial and the sign representation $\sigma$. The trivial bundles may be split off from $\xi$ without affecting anything, since we only study stable tangent structure. Thus, $\xi = m\sigma$ for some $m \geq 0$. The Smith maps go

$$\Omega_S^{D-1}(B\mathbb{Z}_2, (m+1)\sigma) \to \Omega_S^D(B\mathbb{Z}_2, m\sigma). \tag{96}$$

All the others are compositions of these.

If we are interested in bosonic anomalies, then we use oriented cobordism with $S = SO$. In this case, our manifolds have an orientation on $TX \oplus A^* m\sigma$. If $m$ is even, then $m\sigma$ is orientable over $B\mathbb{Z}_2$, so this is equivalent to an orientation of $TX$. Therefore, there is a mod 2 periodicity of the Smith maps, as we have observed:

$$\Omega_O^{D-1} \to \Omega_{SO}^D(B\mathbb{Z}_2) \tag{97}$$

$$\Omega_{SO}^{D-1}(B\mathbb{Z}_2) \to \Omega_O^D. \tag{98}$$

If we are interested in fermionic anomalies, then we use spin cobordism with $S = \text{Spin}$. Now our manifolds have spin structures on $TX \oplus A^* m\sigma$. It turns out that $m\sigma$ admits a spin structure over $B\mathbb{Z}_2$ if $m = 0 \bmod 4$. Thus, there is a mod 4 periodicity of the Smith maps, corresponding to (2):

$$\Omega_{\text{Spin}}^D(B\mathbb{Z}_2) \leftarrow \Omega_{\text{Pin}^-}^{D-1} \leftarrow \Omega_{\text{Spin}^{c/2}}^{D-2} \leftarrow \Omega_{\text{Pin}^+}^{D-3} \leftarrow \Omega_{\text{Spin}}^{D-4}(B\mathbb{Z}_2), \tag{99}$$

where $\text{Spin}^{c/2}$ denotes $\text{Spin}^c$ structure where the $U(1)$ gauge field has holonomies only in the subgroup $\mathbb{Z}_2 < U(1)$. This is the appropriate structure for unitary symmetries $U$ with $U^2 = (-1)^F$.

**Theorem 4.1. Classical Smith Isomorphism** *For $G = \mathbb{Z}_2$, $\xi = 0$, $V = \sigma$, but for any $n$ and structure $S$, the reduced Smith map*

$$\tilde{\Omega}_n^S(B\mathbb{Z}_2) \to \Omega_{n-1}^S(B\mathbb{Z}_2, \sigma) \tag{100}$$

*is an isomorphism. More generally,*

$$\tilde{\Omega}_n^S(B\mathbb{Z}_2, m\sigma) \to \Omega_{n-1}^S(B\mathbb{Z}_2, (m+1)\sigma) \tag{101}$$

*is injective (but not always surjective). Equivalently, the following sequence is exact*

$$\Omega_n^S \to \Omega_n^S(B\mathbb{Z}_2, m\sigma) \to \Omega_{n-1}^S(B\mathbb{Z}_2, (m+1)\sigma), \tag{102}$$

*where the first map is taking an $S$-manifold and considering it as a $\mathbb{Z}_2$-twisted $S$-manifold with trivial $\mathbb{Z}_2$ bundle.*

*Proof.* Let's first prove the first Smith map is surjective. Let us begin with a $(n-1)$-manifold $X$ with $\sigma$-twisted $S$-structure, ie. a $\mathbb{Z}_2$ bundle $A$ and $S$-structure on $TX \oplus A^*\sigma$, where $A^*\sigma$ is a real line bundle. The tangent space of this line bundle is again $TX \oplus A^*\sigma$, where $A$ is pulled back to the total space. We thus obtain an $S$-structure on this open $n$-manifold.

We consider the sphere bundle $Z = \text{Sph}(A^*\sigma)$ obtained by taking the fiber-wise one-point compactification of $A^*\sigma$. This is a compact $n$-manifold with (untwisted) $S$-structure. Further, the zero section, an embedded copy of $X$, is Poincaré dual to a $\mathbb{Z}_2$ bundle $\hat{A}$ on $Z$. Applying the Smith map to $(Z, \hat{A})$ we obtain our original manifold with its twisted $S$-structure.

Now let's prove injectivity in the general twisted case. Suppose $X$ is an $n$-manifold with $\mathbb{Z}_2$ bundle $A$ and $mA^*\sigma$-twisted $S$-structure which is zero under the Smith map. To be precise, we choose a section $s$ of $A^*\sigma$ which is regular at zero so that its zero locus $Y = Y(s, A^*\sigma)$ is an $(n-1)$-submanifold. $Y$ inherits an $(m+1)A^*\sigma$-twisted $S$-structure from $X$.

Since $X$ goes to zero under the Smith map, there is an $n$-manifold $Z$ with $\partial Z = Y$ and to which this twisted $S$-structure extends, with $\mathbb{Z}_2$ bundle $\hat{A}$. We consider the unit interval bundle inside $\hat{A}^*\sigma \to Z$. Call this $\tilde{Z}$. This is an $(n+1)$-manifold with an inclusion

$$\tilde{Y} \hookrightarrow \partial \tilde{Z}, \tag{103}$$

where $\tilde{Y}$ is a tubular neighborhood of $Y$. Thus we can glue $\tilde{Z}$ to $X \times [0,1]$ along $\tilde{Y} \times 1$ to obtain a smooth $n+1$-manifold $W$ one of whose boundary components is $X \hookrightarrow X \times 0$. We denote the union of the other boundary components as $X'$.

Furthermore, $Z \cup (Y \times [0,1]) \subset W$ is Poincaré dual to a $\mathbb{Z}_2$ bundle $\tilde{A}$ over $W$ which restricts to $A$ on $X$ and to nothing on $X'$. By construction, $W$ has an $m\tilde{A}^*\sigma$-twisted $S$-structure extending that of $X$. Thus, $(X, A)$ is bordant to $(X', 0)$ in $\Omega_n^S(BG, m\sigma)$. This means it's zero in reduced bordism, since it is in the image of $\Omega_n^S$. Thus, the map is injective. $\qquad\square$

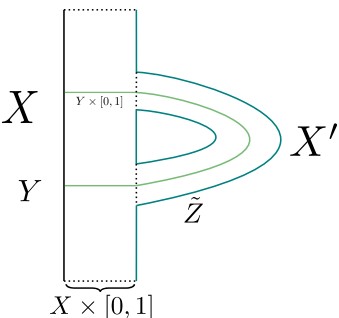

Figure 3: The injectivity proof for the Smith isomorphism theorem in a nutshell: one uses a nullbordism $Z$ of the image of the Smith map $(X,A) \mapsto (Y,A|_Y)$ to construct a bordism $(W,\tilde{A})$, depicted above, of $(X,A)$ to $(X',0)$ (blue), with the latter carrying no $\mathbb{Z}_2$ bundle. Indeed the green curve Poincaré dual to $\tilde{A}$ only meets the boundary of the bordism along $X$.

The injectivity of the reduced (bordism) Smith map is dual to a surjectivity of

$$\Omega_S^{n-1}(B\mathbb{Z}_2, (m+1)\sigma) \to \tilde{\Omega}_S^n(B\mathbb{Z}_2, m\sigma),$$

which means that in the absence of "gravitational" anomalies contained in $\Omega_S^n$, we can trivially gap the theory away from the domain wall and reduce the calculation to a calucation of the anomaly of the induced symmetry on the wall, as we anticipated in Section 2.3.

Here is an important example where the bordism Smith map is not surjective: $\Omega_5^{\text{Pin}^+} = 0$ but $\Omega_4^{\text{Spin}} = \mathbb{Z}$. The problem is that we cannot use our surjectivity trick above because the generator of $\Omega_4^{\text{Spin}}$, the K3 surface, is simply connected. Thus, if we attempt to construct a 5-manifold as above with zero section as the K3 surface, our only choice is $K3 \times S^1$, with nontrivial $\mathbb{Z}_2$ bundle around $S^1$. However, we cannot freely choose the $\mathbb{Z}_2$ bundle for a $\text{Pin}^+$ structure; it has to be the orientation line, but $K3 \times S^1$ is orientable, so its orientation line is trivial, a contradiction.

This is Anderson dual to a failure of injectivity for

$$\mathbb{Z} \oplus \mathbb{Z}_8 = \Omega_{\text{Spin}}^3(B\mathbb{Z}_2) \to \Omega_{\text{Pin}^+}^4 = \mathbb{Z}_{16}, \tag{104}$$

which we saw in an anomaly context in Section 3.1. This is also an example that shows the splitting (70) is non-canonical, since by duality this map must be surjective, while if it commuted with the splitting then the $\mathbb{Z}$ factor would be sent to zero. We do not know a mathematical way to compute this map, but because it is linear we were able to check some well-chosen examples in Section 3.1 and we found it is

$$(k, v_2) \mapsto 2v_2 - k, \tag{105}$$

cf. (43). The noninjectivity in (104) and (43) meant that the anomaly on the domain wall was not determined by the bulk anomaly but on the other hand it did determine the bulk anomaly because we still had surjectivity.

As an example, let us study the hierarchy beginning with $\Omega_{\text{Pin}^+}^0 = \mathbb{Z}_2$. We have, up to $D = 4$,

$$
\begin{array}{ccccccccc}
\mathbb{Z}_2 & \hookrightarrow & \mathbb{Z}_4 & \hookrightarrow & \mathbb{Z}_8 & \hookrightarrow & \mathbb{Z}_8 \oplus \mathbb{Z} & \twoheadrightarrow & \mathbb{Z}_{16} \\
& & \downarrow & & \downarrow & & \downarrow & & \downarrow \\
& & \mathbb{Z}_2 & & \mathbb{Z}_2 & & \mathbb{Z} & & 0
\end{array}
$$

where we have listed the cokernels in the second row, which are the gravitational responses, by the theorem. As we have discussed, the last Smith map is not injective, and requires a computation. From $D = 4$ on we have,

$$\mathbb{Z}_{16} \hookrightarrow\!\!\!\!\to \mathbb{Z}_{16} \hookrightarrow\!\!\!\!\to \mathbb{Z}_{16} \hookrightarrow \mathbb{Z}_{16} \oplus \mathbb{Z}^2 \longrightarrow\!\!\!\!\to \mathbb{Z}_{32} \oplus \mathbb{Z}_2 \hookrightarrow \mathbb{Z}_{64} \oplus \mathbb{Z}_4 \hookrightarrow \cdots$$
$$\downarrow \qquad\quad \downarrow \qquad\quad \downarrow \qquad\qquad \downarrow \qquad\qquad\quad \downarrow$$
$$0 \qquad\quad 0 \qquad\quad \mathbb{Z}^2 \qquad\qquad 0 \qquad\qquad\quad \mathbb{Z}_2^2$$

$$\cdots \hookrightarrow \mathbb{Z}_{128} \oplus \mathbb{Z}_8 \oplus \mathbb{Z}_2 \hookrightarrow\!\!\!\!\to \mathbb{Z}_{128} \oplus \mathbb{Z}_8 \oplus \mathbb{Z}_2 \oplus \mathbb{Z}^3 \longrightarrow\!\!\!\!\to \; ???$$
$$\downarrow \qquad\qquad\qquad \downarrow \qquad\qquad\qquad\quad \downarrow$$
$$\mathbb{Z}_2^3 \qquad\qquad\qquad \mathbb{Z}^3 \qquad\qquad\qquad\quad 0$$

which extends (and corrects) the table of [1]. See [7] for an extensive computation of the gravitational components. Note that where the two 7D and three 11D gravitational Chern-Simons terms appear there is again a kernel of the Smith map, requiring an extra computation, analogous to what we did in Section 3.1. However, we do know that these maps are surjective, since their cokernels are trivial by the theorem. Note also that beginning in $D = 8$ we have another tower analogous to the tower beginning in $D = 0$ above which reflects the ring structure of the bordism group[18]: these $8 + k$ dimensional phases are detected by spacetimes of the form $X_8 \times Y_k$ where $X_8$ is a generator of $\Omega_8^{\text{spin}}$ and $Y_k$ is a test spacetime for a $k$ dimensional phase.

We summarize our calculations using the theorem in the following table[19]. In each row we first list the total group of phases and then below it the cokernel of the incoming Smith map (on the far left is $\Omega_{\text{spin}}^D$, which is the biggest possible cokernel). The Smith maps go down and to the left. Most of these groups can be computed just using the theorem and some low dimensional starting points, see e.g. [40]. To compute the other cokernels and check the results we used Atiyah-Hirzebruch spectral sequence techniques, see [27] for a review.

| $D$ | $U^2 = 1$ | $T^2 = 1$ | $U^2 = (-1)^F$ | $T^2 = (-1)^F$ |
|---|---|---|---|---|
| 1 | $\mathbb{Z}_2 \oplus \mathbb{Z}_2$ | $\mathbb{Z}_2$ | $\mathbb{Z}_4$ | $0$ |
| $\mathbb{Z}_2$ | $\mathbb{Z}_2$ | $\mathbb{Z}_2$ | $\mathbb{Z}_2$ | $0$ |
| 2 | $\mathbb{Z}_2 \oplus \mathbb{Z}_2$ | $\mathbb{Z}_8$ | $0$ | $\mathbb{Z}_2$ |
| $\mathbb{Z}_2$ | $\mathbb{Z}_2$ | $\mathbb{Z}_2$ | $0$ | $\mathbb{Z}_2$ |
| 3 | $\mathbb{Z}_8 \oplus \mathbb{Z}$ | $0$ | $2\mathbb{Z}$ | $\mathbb{Z}_2$ |
| $\mathbb{Z}$ | $\mathbb{Z}$ | $0$ | $2\mathbb{Z}$ | $0$ |
| 4 | $0$ | $0$ | $0$ | $\mathbb{Z}_{16}$ |
| $0$ | $0$ | $0$ | $0$ | $0$ |
| 5 | $0$ | $0$ | $\mathbb{Z}_{16}$ | $0$ |
| $0$ | $0$ | $0$ | $0$ | $0$ |
| 6 | $0$ | $\mathbb{Z}_{16}$ | $0$ | $0$ |
| $0$ | $0$ | $0$ | $0$ | $0$ |

Observe how typically the gravitational terms might not be consistent with the symmetry. The entries where the group of phases equals the cokernel are pure gravitational, and the

---

[18]Our Anderson dual cobordism groups, hence the group of SPT phases, however do not form a ring, because of some degree shifts, or more precisely because $U(1)$ is not a ring.

[19]We would like to thank Miguel Montero for pointing out an error in the $U^2 = (-1)^F$ symmetry class in the first version of this table. It is now consistent with the appendix of [10].

incoming Smith map is zero. For some reason, the hierarchy we discussed in detail above is more interesting than the others—it is also the one where the unitary $U^2 = 1$ symmetry (untwisted spin cobordism) always lines up with the gravitational Chern-Simons terms, so these always give nontrivial phases in this hierarchy. If one had a good understanding of the kernel of the cobordism Smith map, then one could easily compute all of the $\mathbb{Z}_2$ SPT classification, without resorting to spectral sequence techniques.

## 4.5 The Smith Maps for Abelian Groups

In this section we would like to prove some results for general symmetry groups $G$ and their domain walls and junctions, as well as give some more physical interpretations. The most physically interesting results to derive are statements about the kernel of the (bordism) Smith maps, since this kernel is dual to the cokernel of phases or anomalies which are not detected by the domain wall or junction. Our key result is the following:

**Theorem 4.2.** *Let $G = \mathbb{Z}_n$, $\xi$ an arbitrary representation, and $V = V_2$ the two-dimensional real representation given by a $2\pi/n$ rotation of the plane. The following sequence is exact*

$$\tilde{\Omega}_n^S(B\mathbb{Z}, \xi) \to \tilde{\Omega}_n^S(B\mathbb{Z}_n, \xi) \to \Omega_{n-2}^S(B\mathbb{Z}_n, \xi \oplus V_2), \tag{106}$$

*where the first map takes a $\mathbb{Z}$ gauge field and forms its quotient to give a $\mathbb{Z}_n$ gauge field, and the second map is the Smith map based on $V_2$.*

*Proof.* First we must show the image of the first map is in the kernel of the second map. This is because the unit circle bundle inside $A^*V_2$ has Chern class $e(A^*V_2)$, which is zero if $A$ lifts to a $\mathbb{Z}$ gauge field, hence $V_2$ admits a nonvanishing section. (Note for higher dimensional bundles having a vanishing Euler class is not sufficient to guarantee a nonvanishing section.)

Conversely, we begin with an $(X,A)$ in the kernel of the second map and we want to show it is in the image of the first map. We follow the construction in the proof of Theorem 4.1, which gives us a bordism from $(X,A)$ to $(X',A')$ where $e(A'^*V_2) = 0$. This means $A'$ has a lift to a $\mathbb{Z}$ bundle since $e(A'^*V_2)$ is the Bockstein of $A'$. Thus, $(X',A')$ and hence $(X,A)$ is in the image of the first map. $\qquad\square$

This kernel is dual to a cokernel of phases (or anomalies) which are not distinguished by the symmetric junction. These phases are determined by their image in $\tilde{\Omega}_S^n(B\mathbb{Z}, \xi)$. This group actually has a simple characterization. These are the terms which are linear in $A$, in the sense that

$$\tilde{\Omega}_S^n(B\mathbb{Z}, \xi) = H^1(B\mathbb{Z}, \Omega_S^{n-1}), \tag{107}$$

which may be derived from the Atiyah-Hirzebruch spectral sequence. Intuitively, such a linear term implies that the domain wall carries a fermionic phase which is nontrivial without any symmetry.

For example, with $G = \mathbb{Z}_2$, we have $2 + 1$ dimensional phases representing a nontrivial element of

$$H^1(B\mathbb{Z}_2, \Omega_{\text{spin}}^2) = \mathbb{Z}_2, \tag{108}$$

characterized by the domain wall carrying a Kitaev wire. One might worry about this leading to a gravitational anomaly on the junction but actually it works out because the $\mathbb{Z}_2$ junction has two domain walls coming in. The $\mathbb{Z}_2$ junction gives us the map

$$\mathbb{Z}_4 = \Omega_{\text{spin}}^1(B\mathbb{Z}_2, \sigma \oplus \sigma) \to \Omega_{\text{spin}}^3(B\mathbb{Z}_2) = \mathbb{Z}_8, \tag{109}$$

and we see this is consistent with the $\mathbb{Z}_2$ cokernel above. Note that since $V_2 = \sigma \oplus \sigma$ for $\mathbb{Z}_2$, this map is the composition of the two Smith maps for $\sigma$:

$$\Omega_{\text{spin}}^1(B\mathbb{Z}_2, \sigma \oplus \sigma) \to \Omega_{\text{spin}}^2(B\mathbb{Z}_2, \sigma) \to \Omega_{\text{spin}}^3(B\mathbb{Z}_2) \tag{110}$$

$$\mathbb{Z}_4 \to \mathbb{Z}_8 \to \mathbb{Z}_8 \oplus \mathbb{Z}. \tag{111}$$

As expected from Theorem 4.1, the second is an isomorphism onto the reduced piece while the first has cokernel $\Omega^2_{\text{spin}} = \mathbb{Z}_2$.

Another interesting example is with $n = 4k$, we have

$$\Omega^3_{\text{Spin}}(B\mathbb{Z}_n) = \mathbb{Z}_{2n} \oplus \mathbb{Z}_2. \tag{112}$$

The first piece describes Gu-Wen-Freed phases [31, 32], while the second piece is the phase with the Kitaev wire on the domain wall. We can detect this wire using the one-dimensional representation $\sigma$, we find the Smith map is

$$\Omega^2_{\text{Spin}}(B\mathbb{Z}_n, \sigma) = \mathbb{Z}_4 \oplus \mathbb{Z}_2 \to \Omega^3_{\text{Spin}}(B\mathbb{Z}_n), \tag{113}$$

with the first factor surjecting onto the $\mathbb{Z}_2$ factor of $\Omega^3_{\text{Spin}}(B\mathbb{Z}_n)$ and the second going to zero. Meanwhile, under the $V = V_2$ Smith map,

$$\Omega^1_{\text{Spin}}(B\mathbb{Z}_n, V_2) = \mathbb{Z}_{2n} \to \Omega^3_{\text{Spin}}(B\mathbb{Z}_n) \tag{114}$$

surjects onto the $\mathbb{Z}_{2n}$ piece, leaving the cokernel $\mathbb{Z}_2$ as before.

Further, for $n$ odd, $H^1(B\mathbb{Z}_2, \Omega^{n-1}_S) = 0$ for $S = Spin$ or $SO$, since for these $\Omega^{n-1}_S$ is a product of $\mathbb{Z}_2$'s and possibly $\mathbb{Z}$'s in dimensions $4k-1$ from gravitational Chern-Simons terms [7]. Thus for $n$ odd, the Smith homomorphism relevant for $\mathbb{Z}_n$ anomalies based on the two-dimensional representation is injective, and the $\mathbb{Z}_n$ anomaly is uniquely determined by the junction.

Finally we note that because the theorem is proved for arbitary structure $S$, we can apply it to product groups $G = \mathbb{Z}_n \times H$, where the $H$ gauge field and possible $H$-twisted spin structure or orientation are considered part of the tangent structure $S$. We can thus bootstrap these theorems to results about any finite abelian group, for which we find the anomaly is characterized by splitting $G = \mathbb{Z}_n \times H$ for each cyclic factor $\mathbb{Z}_n$, looking at the $H$ anomaly on the $\mathbb{Z}_n$ domain wall, and looking at the $G$ anomaly on the codimension-2 $\mathbb{Z}_n$ junction.

## 5 Discussion

Other dimensional hierarchies of topological phases have been considered before, e.g. in free systems in [26]. In [25], the authors described a dimensional reduction procedure which applies to free fermion phases modified by strong interactions. While apparently quite similar to our example in Section 3.1 the overall picture of phases which appeared in their "Bott spiral", which includes all the symmetry classes in the 10-fold way, has a very much more regular structure than what we found in Section 4.4. It would be very interesting to relate the two pictures and lead to a better understanding of interacting SPT phases.

There is an apparent similarity between our dimensional reduction procedures and the decorated domain wall methods [11,15,16,57]. One should consider dimensional reduction as a probe of a phase or anomaly, while decorated domain walls are a method to construct them. They have to be consistent. For instance, if one gives a decorated domain wall construction of a $2+1$ dimensional SPT phase by placing a Kitaev wire on the domain wall, then this phase is the image of the Smith map of the sign representation applied to the $1+1$ dimensional Kitaev phase.

However, in general there are many possible domain wall decorations not described by Smith maps and they must satisfy some complicated consistency relations, not all of which are known (see [27] for a review). These form the differentials of the Atiyah-Hirzebruch spectral

sequence (AHSS), a rather abstract algebraic object. It's especially nice to have a geometric understanding of these consistency conditions, since this usually comes along with some physical intuition. We expect this to be possible because so far all of the known consistency conditions have some degree of understanding along these lines.

More precisely, the decorated domain wall construction begins by considering elements

$$\omega \in H^k(BG, \Omega_S^{D-k}) \tag{115}$$

in the $E_2$ page of the AHSS, which describes a decoration of certain codimension-$k$ defects of a $G$-gauge field with $D-k$ dimensional fermionic phases in $\Omega_S^{D-k}$. We expect that when $\omega$ is the Euler class of a rank-$k$ representation $V$, then it can be extended to a fully consistent decorated domain wall construction of a fermionic phase which is in the image of a Smith map based on $V$. One needs to be careful about the twists to obtain a precise statement. If something like this is true, it would place strong geometric constraints on the AHSS differentials, and be very interesting. This might also lead to a deeper understanding of Theorem 4.2, which appears to be a statement about the "extension problem" of the AHSS in this context, cf. (107).

With regards to crystalline symmetries, there is an even simpler procedure to reduce to the domain wall, first described by [42, 43]. For example, suppose we have a unitary reflection symmetry across some hyperplane. We disorder the system on one side of the hyperplane by some interaction, and then add the reflection-conjugated interaction to the other side of the hyperplane to obtain a system localized to the hyperplane but still with the unitary symmetry, although now it acts internally!

One may wonder if the anomaly on the hyperplane can be understood in terms of the original anomaly and it turns out it can. By the crystalline equivalence principle of [41], we can identify the reflection SPT with an associated time reversal SPT (see also [52]). We see that reflection and time reversal are related by $CPT$. Then applying our reduction, we obtain a unitary internal symmetry on the hyperplane, which is the same as the unitary above, since $(CPT)^2 = 1$. More generally, if one examines the necessary twists in the crystalline equivalence principle, one finds they exactly cancel the twists in the Smith maps, so the crystalline Smith map *does not* change the type of symmetry enjoyed on the Wyckoff position, although it becomes internal. See also [4].

If one takes a subspace larger than a Wyckoff position, e.g. a coordinate plane in $\mathbb{R}^3$, where our unitary $\mathbb{Z}_2$ acts as parity symmetry $x, y, z \mapsto -x, -y, -z$, the symmetry goes from an orientation-reversing symmetry to an orientation-preserving rotation on the plane, which is the crystalline analog of the 2-fold periodic structure, and becomes a 4-fold periodic structure when fermions are carefully accounted for. See [12–14]. When one uses equivariant homology all twists in the dimensional reduction disappear, indicating that in the crystalline setting, they are just due to Poincaré duality.

However, if our symmetry acts internally in the lattice model, we do have to break the symmetry to form the domain wall, and in this case it is not clear how we should define the symmetry on the wall. We leave this interesting question to future work.

# Acknowledgments

I.H is supported in part by the Clore Foundation, the I-CORE program of Planning and Budgeting Committee (grant number 1937/12), the US-Israel Binational Science Foundation, GIF and the ISF Center of Excellence. Z.K is supported in part by the Simons Foundation grant 488657 (Simons Collaboration on the Non-Perturbative Bootstrap). R.T is supported by the Zuckerman STEM Leadership Program and the NSF GRFP Grant Number DGE 1752814.

# A Conventions for the action of $C, P, T$

We work out the action of $C, P, T$ on the minimal possible fermion representation in $2+1$, $1+1$ and $0+1$ dimensions. The more general cases are treated briefly since the conclusions remain the same.

## A.1 $2+1$ Dimensions

We take the sigma matrices to be

$$\sigma^1 = \begin{pmatrix} 0 & 1 \\ 1 & 0 \end{pmatrix}, \quad \sigma^2 = \begin{pmatrix} 0 & -i \\ i & 0 \end{pmatrix}, \quad \sigma^3 = \begin{pmatrix} 1 & 0 \\ 0 & -1 \end{pmatrix}. \tag{116}$$

They satisfy the usual relations

$$\{\sigma^i, \sigma^j\} = 2\delta^{ij}, \qquad [\sigma^i, \sigma^j] = 2i\epsilon^{ijk}\sigma^k, \tag{117}$$

where $\epsilon^{123} = 1$.

We will need to adapt these matrices for the Lorentzian signature in $2+1$ dimensions that we are going to use. We will denote the corresponding matrices by $\gamma^{0,1,2}$ and we will take the signature to be $(-,+,+)$ and the metric is denoted by $\eta^{\mu\nu}$ with $\mu, \nu = 0, 1, 2$.

$$\gamma^0 = i\sigma^2, \qquad \gamma^1 = \sigma^1, \quad \gamma^2 = \sigma^3. \tag{118}$$

These satisfy

$$\{\gamma^\mu, \gamma^\nu\} = 2\eta^{\mu\nu}, \qquad [\gamma^\mu, \gamma^\nu] = 2\epsilon^{\mu\nu\rho}\gamma_\rho. \tag{119}$$

The generators of the Lorentz group $SO(2,1)$ are just the $[\gamma^\mu, \gamma^\nu]$ such that on a two-dimensional spinor $\lambda_\alpha$ (at the origin) the Lorentz transformations act as $\lambda' = e^{\frac{1}{2}[\gamma^\mu, \gamma^\nu]\Theta_{\mu\nu}}\lambda$, where the $\Theta_{\mu\nu}$ parameterize boosts and rotations. The $\Theta_{\mu\nu}$ are real anti-symmetric matrices. We can re-write the transformation as $\lambda' = e^{\gamma^\rho \Xi_\rho}\lambda$ with $\Xi_\rho = \epsilon^{\mu\nu}{}_\rho\Theta_{\mu\nu}$.

We can impose a Majorana condition on $\lambda$ by noting that all the $\gamma$ matrices are real and hence we can take $\lambda_\alpha$ to be real. We define $\bar\lambda \equiv \lambda^T\gamma^0$ and see that it transforms as

$$\bar\lambda \to \lambda^T(e^{\gamma^\rho \Xi_\rho})^T\gamma^0 = \lambda^T\gamma^0 e^{-\gamma^\rho \Xi_\rho} = \bar\lambda e^{-\gamma^\rho \Xi_\rho}. \tag{120}$$

As a result, $\bar\lambda\lambda$ is invariant and $\bar\lambda\gamma^\mu\partial_\mu\lambda$ is likewise an invariant. Note that

$$(\bar\lambda\gamma^\mu\partial_\mu\lambda)^\dagger = -\partial_\mu\lambda^T(\gamma^\mu)^T\gamma^0\lambda = \partial_\mu\lambda^T\gamma^0\gamma^\mu\lambda = \partial_\mu\bar\lambda\gamma^\mu\lambda,$$

and hence integrating by parts we get a minus sign and hence we need to put an $i$ in front of the kinetic term as well as in front of the mass term

$$\int d^3x \; i\bar\lambda\gamma^\mu\partial_\mu\lambda + iM\bar\lambda\lambda. \tag{121}$$

Time reversal symmetry and parity act as follows:

$$T : \lambda(x^0, x^1, x^2) \to \pm\gamma^0\lambda(-x^0, x^1, x^2), \tag{122}$$

$$P : \lambda(x^0, x^1, x^2) \to \pm\gamma^1\lambda(x^0, -x^1, x^2). \tag{123}$$

The signs are uncorrelated and arbitrary in principle. But we will take the two signs in $P, T$ to be always correlated.

For the mass term: $T(\bar{\lambda}\lambda) = -\lambda^T \gamma^0 \gamma^0 \gamma^0 \lambda = \bar{\lambda}\lambda$. Taking into account the factor of $i$ in the mass term and that $T$ is anti-linear, we find that under time reversal symmetry $M \to -M$. Applying parity, $P(\bar{\lambda}\lambda) = \lambda^T \gamma^1 \gamma^0 \gamma^1 \lambda = -\bar{\lambda}\lambda$ and parity is a linear operator so the mass is again seen to be odd under parity.

The action on the kinetic term is $T(\bar{\lambda}\gamma^\mu \partial_\mu \lambda) = -\bar{\lambda}\gamma^0 \partial_0 \lambda - \bar{\lambda}\gamma^i \partial_i \lambda = -\bar{\lambda}\gamma^\mu \partial_\mu \lambda$ and again together with the factor of $i$ in front of the kinetic term we find that the kinetic term is time reversal even. Similarly, it is parity even.

Three important properties that we immediately recognize are

$$
\begin{aligned}
(CPT)^2 &= 1 \,, \\
T^2 &= (-1)^F \,, \\
T \cdot CPT &= (-)^F CPT \cdot T \,.
\end{aligned}
\tag{124}
$$

It is important to note that because $T$ is anti-linear all the three relations in (124) are invariant under multiplying $T$ by $i$. For the same reason the first relation is also invariant under multiplying $P$ by $i$. In the last relation we can add an arbitrary $c$ number phase but the existence of $(-1)^F$ is invariant.

## A.2   $1+1$ **Dimensions**

The signature is chosen to be $(-,+)$ and in terms of the sigma matrices (116) we have as before

$$
\gamma^0 = i\sigma^2 \,, \gamma^1 = \sigma^1 \,.
\tag{125}
$$

The boost transformations are given by $e^{\beta[\gamma^0, \gamma^1]}$ with real $\beta$. Therefore the representation is reducible and we can call the boost eigenstates as $\lambda_+$ and $\lambda_-$. The non-chiral fermion is given simply by

$$
\lambda = \begin{pmatrix} \lambda_+ \\ \lambda_- \end{pmatrix} \,.
\tag{126}
$$

The kinetic term and mass term are as in $2+1$ dimensions (except that now the index $\mu$ ranges over 1,2)

$$
\mathcal{L} = i\lambda^T \gamma^0 \gamma^\mu \partial_\mu \lambda + iM \lambda^T \gamma^0 \lambda \,.
\tag{127}
$$

For now let us assume that charge conjugation symmetry acts trivially, time reversal symmetry acts as before and parity acts as before:

$$
T : \lambda(x^0, x^1) \to \pm \gamma^0 \lambda(-x^0, x^1) \,,
\tag{128}
$$

and we note that the mass term is odd under time reversal symmetry. In particular, as expected time reversal symmetry acts by exchanging fermions that are moving to the left with fermions that are moving to the right.

We find again the relations

$$
\begin{aligned}
(CPT)^2 &= 1 \,, \\
T^2 &= (-1)^F \,, \\
T \cdot CPT &= (-)^F CPT \cdot T \,.
\end{aligned}
\tag{129}
$$

For the massless fermion $\lambda$ however we do not need to assume that charge conjugation symmetry acts trivially. Up to overall conjugation by $(-1)^F$ there is one more nontrivial choice, where

$$
C : \lambda \to \sigma^3 \lambda \,,
\tag{130}
$$

namely, it acts like fermion number only on $\lambda_-$ but not on $\lambda_+$. Sometimes it would be denoted by $(-1)^{F_L}$. This is a chiral $\mathbb{Z}_2$ symmetry. The transformation (130) commutes with boosts and hence with the Poincaré group. However, it does not commute with time reversal symmetry or parity. This charge conjugation symmetry likewise forbids a mass term.

With the choice (130) we find the relations

$$
\begin{aligned}
C^2 &= 1 \,, \\
CT &= (-1)^F TC \,, \\
(CPT)^2 &= 1 \,, \\
T^2 &= (-1)^F \,, \\
T \cdot CPT &= CPT \cdot T \,.
\end{aligned}
\tag{131}
$$

Note that the difference in the last equation is due to the fact that this $CPT$ is not the canonical one.

### A.3   $0+1$ **Dimensions**

Here the minimal fermion is just a single component fermion $\lambda$. The Lagrangian is

$$
\mathcal{L} = i\lambda \frac{d}{dt}\lambda \,.
\tag{132}
$$

This theory is however trivial since upon quantization we have one Hermitian operator $\lambda$ that also satisfies $\lambda, \lambda = 1$ and so the Hilbert space consists of only one state.

Now let us consider a collection of such fermions

$$
\mathcal{L} = i\sum_{I=1}^{N} \lambda^I \frac{d}{dt}\lambda^I \,.
\tag{133}
$$

The Hamiltonian again vanishes identically and the operators $\lambda^I$ satisfy the Clifford algebra

$$
\{\lambda^I, \lambda^J\} = \delta^{IJ} \,,
\tag{134}
$$

in addition, the $\lambda^I$ are Hermitian. It is well known that the construction proceeds slightly differently for even and odd $N$. For even $N$ we combine the fermions into pairs (in an arbitrary fashion)

$$
\psi^k = \lambda^k + i\lambda^{k+N/2} \,, \quad k = 1, ..., N/2
\tag{135}
$$

and we have that $\{\psi^k, \psi^{k'}\} = 0$, $\{\psi^k, \bar{\psi}^{k'}\} = 2\delta^{k,k'}$ which means that we have a $2^{N/2}$ dimensional Hilbert space isomorphic to $\bigotimes_{k=1}^{N/2}|s_k\rangle$ with $s_k = \pm 1$. The $\psi^k$ acts only on the kth spin such that

$$
\psi^k = \sqrt{2}\begin{pmatrix} 0 & 1 \\ 0 & 0 \end{pmatrix}, \qquad \bar{\psi}^k = \sqrt{2}\begin{pmatrix} 0 & 0 \\ 1 & 0 \end{pmatrix}
\tag{136}
$$

and the different $\psi$'s all anti-commute otherwise. Now we need to define a time-reversal operation. We can define it in each of the $N/2$ blocks.

The theory has charge conjugation symmetry $C$ implemented on the Hilbert space by the matrix $\begin{pmatrix} 0 & 1 \\ 1 & 0 \end{pmatrix}$. It is instructive to consider the $U(1)^{N/2}$ symmetry inside the $SO(N)$ symmetry and require that charge conjugation symmetry reverses those charges

$$
Ce^{i\alpha Q} = e^{-i\alpha Q}C \,.
\tag{137}
$$

Then in this case it is known that if we choose the $U(1)$ charges to be integral the algebra of $C, Q$ is centrally extended while if we allow for half-integral charges then the central extension can be removed. We can readily see that as an operator

$$C^2 = 1 , \qquad C\psi C = \psi^\dagger , \qquad C\psi^\dagger C = \psi . \tag{138}$$

In particular, the fundamental representation goes to the anti-fundamental representation. It is easy to verify that this leaves the action invariant.

We can take $Q = \psi\psi^\dagger - 1/2$ so that the equations above are all mutually consistent. If we do not include the factor of $1/2$, we get some central extension of the $C, Q$ algebra, which reflects the well known $O(2)$ anomaly in QM.

Similarly time reversal reverses those $U(1)$ charges. So this implies that

$$e^{i\alpha Q}T = Te^{i\alpha Q} . \tag{139}$$

Note that because of the $i$ in the exponent and the anti-linearity of $T$, $TQ = -QT$ is equivalent to the above.

We can define $T$ as the composition of complex conjugation and $C$ above, which again would lead to a central extension if the $U(1)$ charges are chose to be integral. Note that with this definition $T^2 = 1$.

Interestingly, we can a priori also choose a different action of $T$, which is obtained by a composition of the above-chosen time reversal symmetry and a rotation by $\pi$. This leads to time reversal symmetry acting by the combination of complex conjugation and the matrix $\begin{pmatrix} 0 & -1 \\ 1 & 0 \end{pmatrix}$. Now we have $T^2 = -1$ and taking into account that we have $N/2$ blocks we find $T^2 = (-1)^{N/2}$. For an anti-unitary operator, $T^2 = -1$ cannot be converted to $T^2 = 1$ by multiplying the operator with $i$.

We can consider with the above conventions the transformation $CT$. It always satisfies $(CT)^2 = 1$. This is the analog of $CPT$. We can also think about $CT$ as fermion number because (if it is nontrivial, then) it acts like $diag(-1, 1)$ which is the same as multiplying all the fermions by a minus sign. This is why it is an unbreakable symmetry and it is the correct analog of CPT.

It is nice to note that with the above choice of $C$ and $T$ such that $CT$ is fermion number, we find that

$$CT \cdot T = -T \cdot CT \tag{140}$$

and taking into account that we have $N$ fermions

$$CT \cdot T = (-)^{N/2} T \cdot CT , \tag{141}$$

which is analogous to the result in higher dimensions if we think of $(-1)^{N/2}$ as $(-1)^F$.

# B  Analytic Continuation and CPT

## B.1  Analytic Continuation

We study correlation functions

$$\langle \phi_1(x_1, z_1) \cdots \phi_n(x_n, z_n) \rangle \tag{142}$$

for complex times $z_j = t_j + i\tau_j$. We can rewrite this in terms of a real time correlator

$$\langle e^{-\tau_1 H} \phi_1(x_1, t_1) e^{(\tau_1 - \tau_2)H} \phi_2(x_2, t_2) \cdots \rangle \tag{143}$$

We see for these correlation functions to be finite they must be imaginary-time-ordered, meaning that the $\tau$'s are increasing

$$\tau_1 < \tau_2 < \cdots \tag{144}$$

so that all the exponential factors come with a negative factor. Note that we don't worry about the first exponential factor because it is absorbed by the ground state on the left.

## B.2 Reflection Positivity

The first identity we can assert considering these correlation functions is given by unitarity, for which

$$\phi(x,z)^\dagger = (e^{izH}\phi(x,0)e^{-izH})^\dagger \tag{145}$$

$$= e^{iz^*H}\phi(x,0)^\dagger e^{-iz^*H}. \tag{146}$$

Thus, if $\phi(x,0)$ is a Hermitian operator, then

$$\phi(x,z)^\dagger = \phi(x,z^*). \tag{147}$$

We find therefore

$$\langle\phi(x,z^*)\phi(x,z)\rangle > 0, \tag{148}$$

and so on for more complicated operator insertions. Let us note that for $z = i\tau$, this reads

$$\langle\phi(x,-i\tau)\phi(x,i\tau)\rangle > 0, \tag{149}$$

which is automatically imaginary-time-ordered as long as $\tau > 0$ (which is required for the states $\phi(x,z)|0\rangle$ to exist), and is moreover reflection symmetric under $\tau \mapsto -\tau$, hence the term reflection positivity.

## B.3 Time Reversal

Now let us suppose that our theory has an anti-unitary symmetry $T$ with

$$HT = TH. \tag{150}$$

Any such symmetry is regarded as a time-reversal symmetry because

$$T^{-1}e^{itH}T = e^{-itH} \tag{151}$$

is functionally the same as $t \mapsto -t$. However, for imaginary time, it takes a bit more care to see what it should do.

First, let us note that anti-unitarity of $T$ means that for any pair of Hilbert space states,

$$\langle a|b\rangle = \langle Ta|Tb\rangle^* = \langle Tb|Ta\rangle. \tag{152}$$

Now we consider the imaginary-time-ordered correlation function

$$\langle\phi_1(z_1)\phi_2(z_2)\rangle, \tag{153}$$

ie. $\tau_1 < \tau_2$, where we have suppressed the position coordinates because they are irrelevant at this stage of the discussion. We can write this as $\langle 0|\phi_1(z_1)\phi_2(z_2)0\rangle$ and apply $T$ to obtain

$$\langle\phi_1(z_1)\phi_2(z_2)\rangle = \langle T^{-1}\phi_2(z_2)^\dagger T T^{-1}\phi_1(z_1)^\dagger T\rangle = \langle(T^{-1}\phi_2 T)(-z_2)(T^{-1}\phi_1 T)(-z_1)\rangle. \tag{154}$$

We observe that the final result is still imaginary-time-ordered, as $-\tau_2 < -\tau_1$! Thus, time reversal acts geometrically in analytic continuation as $z \mapsto -z$.

### B.4 The $CPT$ Algebra

$CPT$ symmetry arises in Euclidean signature from the analytic continuation of a boost along a coordinate $x$ to parameter $i\pi$, for which it becomes a rotation in the $x$-$\tau$ plane. This operator is: 1) a symmetry just like an ordinary boost, 2) anti-unitary because it flips time. Note that a normal boost does not have a well-defined unitary property because it doesn't fix a time slice. We can call this operator $CPT$ because we can easily decompose it to a $P$ part, due to the space coordinate flip, a $T$ part due to the time coordinate flip, and a unitary internal $C$ part which is whatever is needed to complete the composition of the operator, in fact you can define $C = (CPT) \cdot T^{-1} \cdot P^{-1}$, where $CPT$ is the operator constructed above and $P$ and $T$ are any reflection and time reversal operators (which need not be symmetries), respectively.

Now that we have this representation, we want to argue for $(CPT)^2 = 1$.[20] In cases where the theory can be analytically continued to imaginary time, one can consider the $i\pi$ boost as a $\pi$ rotation. A $\pi$ rotation on operators depends on their spin, where a factor of $(-1)^s$ accompanies the obvious coordinates rotation, i.e., even integer spin operators get a $(+1)$ factor, odd integer spin operators get $(-1)$ and half integer spin operators get a $(\pm i)$ factor. Thus, as is well known, under two $\pi$ rotations, or a $2\pi$ rotation, one gets a $(-1)^F$ factor. However, since $T$ complex conjugates as well as reflects time, the $(\pm i)$ factor that the half-integer spin operators get after the first $CPT$ operation is complex conjugated by the second $CPT$ operation and cancels the second factor and therefore in total one gets $(CPT)^2 = 1$.

In order to address the commutation of $T$ and $CPT$ we use similar arguments. $CPT$ by itself is, in the Euclidean analytic continuation, an anti-unitary $\pi$ rotation and since $T$ is anti-unitary, if we operate with $T$ after $CPT$ it reverses the sign of the $\pm i$ factor the half-integer operators received, compared to the other way around, when you operate first with $T$ and only then with $CPT$. Thus, in total, one gets $T \cdot CPT = (-1)^F CPT \cdot T$.

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
