# Peer review of "Anomaly Matching in the Symmetry Broken Phase: Domain Walls, CPT, and the Smith Isomorphism"

_SciPost Physics, doi:SciPost Phys. 8, 062 (2020)_

## Round 2 · Referee Report · Anonymous · 2020-2-21

Strengths

1- This paper provides a useful way to evaluate ‘t Hooft anomalies of higher-dimensional theories from those of lower-dimensional theories.

2- For such a technique, a careful identification of the symmetry of domain walls in spontaneously broken phase is important. They point out the importance of the canonical CPT symmetry.

3- This logic can be reversed, and the knowledge of anomaly for higher-dimensional field theories is helpful to obtain light excitations on the domain wall, when the anomaly in higher-dimensional QFT is matched by SSB of discrete symmetry. This study has an interesting application for this purpose, too.

Weaknesses

Nothing in particular.

Report

This paper focuses on how anomaly matching can be satisfied when discrete symmetry is spontaneously broken. When discrete symmetry is spontaneously broken, one can consider the domain wall which connects different domains. When the anomaly is present, the domain-wall physics is constrained by that anomaly and should contain light excitations similar to that of Jackiw-Rebbi mechanism.

In order to get a rigorous understanding on domain-wall physics, the authors studied carefully about the symmetry on the wall. Especially, they consider the following question first: If $Z_2$ symmetry is spontaneously broken on the bulk, does the domain wall have unbroken $Z_2$ symmetry? The answer turns out to be slightly nontrivial: The combination of the broken $Z_2$ and $CPT$ gives a good symmetry on the wall.

Then, what happens if the bulk theory has an ‘t Hooft anomaly for such $Z_2$ symmetry? To satisfy the anomaly matching, the domain wall theory should support a nontrivial QFT with lower-dimensional ‘t Hooft anomaly. This is partly known in previous studies, but they make a more rigorous connection between the original anomaly and the anomaly on the wall. This allows us to reconstruct the anomaly of higher-dimensional QFT from anomalies of lower-dimensional QFTs, which are more tractable in many cases.

These are very interesting and useful results in the study of QFTs, so I strongly recommend its publication.

I only have one question. The authors obtained a lower-dimensional QFT by considering the domain wall with the frustrated boundary condition along a line or a large circle. Instead, we can obtain another lower-dimensional QFT with the same b.c. along a small circle and integrating out non-zero KK modes. Do we have a similar subtlety of the symmetry and anomaly related to the combination of CPT invariance also in such cases?

Requested changes

No changes are needed

  • validity: top
  • significance: top
  • originality: top
  • clarity: top
  • formatting: perfect
  • grammar: perfect

Author:  Ryan Thorngren  on 2020-04-07  [id 792]

(in reply to Report 1 on 2020-02-21)
Category:
answer to question

Dear Colleague,

Thank you for your report.

Regarding your question, we also expect that this story will be in some part repeated in the study of defects. For instance, if U is an ordinary Z/2 symmetry, that is---unitary and squaring to the identity, it is well-known that in general there is no induced Z/2 action on the defect Hilbert space, since one may have to resolve crossings between the U insertions. For instance, for bosons in 1+1D, it is known that the Z/2 anomaly is captured by the nontrivial crossing relations of the U lines.

However it appears that one can indeed devise an action of U*CPT on the defect Hilbert space, and the associated anomaly in 0+1D is the Kramers degeneracy protected by this anti-unitary symmetry. (This seems to reflect the negative Frobenius-Schur indicator of the Z/2 vortex in the 2+1D theory.)

From the perspective of the Smith isomorphism, something like this probably happens in general dimensions, at least in this symmetry class. Indeed, in the first two paragraphs of the proof of theorem 4.1 we show that compactification with a U-twist is inverse to the Smith homomorphism (an isomorphism in this case).

On the other hand, this is the direction of the proof which fails in the other symmetry classes. Indeed, for time reversal, we don't know a good way to define the defect Hilbert space, but even for the other unitary Z/2 class with U^2 = (-1)^F something seems amiss.

best wishes,
Itamar Hason, Zohar Komargodski, and Ryan Thorngren

---

## Editorial Decision

published